# Clearance of protein aggregates during cell division

**Shoukang Du[1,2], Yuhan Wang[1], Bowen Chen[1], Shuangshuang Xie[1], Kuan Yoow Chan[1,2], David C Hay[3], Ting Gang Chew[1,2,4,5]***

[1]The Zhejiang University-University of Edinburgh Institute, Zhejiang University School of Medicine, Zhejiang University, Hangzhou, China; [2]College of Medicine and Veterinary Medicine, The University of Edinburgh, Edinburgh, United Kingdom; [3]Centre for Regenerative Medicine, Institute for Regeneration and Repair, University of Edinburgh, Edinburgh, United Kingdom; [4]Department of Cardiology of the Second Affiliated Hospital, Zhejiang University School of Medicine, Zhejiang University, Hangzhou, China; [5]State Key Laboratory of Biobased Transportation Fuel Technology, Zhejiang University, Hangzhou, China

**\*For correspondence:**
tinggchew@intl.zju.edu.cn

## eLife Assessment

How misfolded proteins are segregated and cleared is a significant question in cell biology, since clearance of these aggregates can protect against pathologies that may otherwise arise. The authors discover a cell cycle stage-dependent clearing mechanism that involves the ER chaperone BiP, the proteosome, and CDK inactivation, but is curiously independent of the anaphase promoting complex (APC). These are **valuable** and interesting new observations, and the evidence supporting these claims is **solid**.

**Abstract** Protein aggregates are spatially organized and regulated in cells to prevent the deleterious effects of proteostatic stress. Misfolding of proteins in the endoplasmic reticulum (ER) results in aggregate formation, but how the aggregates are processed, especially during cell division is not well understood. Here, we induced proteostatic stress and protein aggregation using a proteostasis reporter, which is prone to misfolding and aggregation in the ER. Unexpectedly, we detected solid-like protein aggregates deposited mainly in the nucleus and surrounded by the ER membrane. The membrane-bound aggregates were then cleared as cells progressed through mitosis and cytokinesis. Aggregate clearance depended on Hsp70 family chaperones in the ER, particularly BiP, and proteasomal activity. The clearance culminated at mitotic exit and required cyclin-dependent kinase 1 (Cdk1) inactivation but was independent of the anaphase-promoting complex (APC/C). The ER reorganization that is active during mitosis and cytokinesis was required for the aggregate clearance. Thus, dividing cells reorganize the ER networks to allow BiP to clear the protein aggregates to maintain proteostasis in the newly divided cells.

## Introduction

Nascent polypeptides fold into three-dimensional structures to perform their biological functions (*Hipp et al., 2019*). Misfolded proteins tend to have their hydrophobic residues exposed and result in aggregation of proteins (*Hartl et al., 2011*). Intramolecular beta-sheets in some proteins are also structural elements prone to protein aggregation (*Tyedmers et al., 2010*). Protein misfolding and aggregation not only disrupt the native function of the protein but also could interfere with other proteins' functions by co-aggregation (*Olzscha et al., 2011*; *Woerner et al., 2016*). This challenges

proteome homeostasis (proteostasis) and causes proteostatic stress in cells, which is a main driver of cellular aging and neurodegenerative disorders (*Balch et al., 2008*; *Labbadia and Morimoto, 2015*).

Cells evolve several protein quality control (PQC) systems to maintain their proteostasis in which components of PQC systems facilitate refolding, degradation, and spatial deposition of misfolded protein aggregates (*Jayaraj et al., 2020*). Molecular chaperones are key players in the PQC system. They are involved to fold nascent polypeptides, target misfolded proteins for degradation by the ubiquitin-proteasome system or autophagy, and promote solubilization of protein aggregates (*Tyedmers et al., 2010*). Eukaryotic cells confine misfolded proteins and smaller aggregates into distinct cellular deposition sites, which reduce the reactivity of these harmful species to the proteome (*Kaganovich et al., 2008*). These deposition sites include aggresomes or aggresome-like induced structures locating near centrosomes, INQ (intranuclear quality control compartment) residing in the nucleus, cytosolic CytoQ, and the perivacuolar IPOD (insoluble protein deposit) (*Kaganovich et al., 2008*). Confinement of protein aggregates in a deposition site facilitates retention of aggregates in short-lived cells (mother cells) during asymmetric cell division (*Aguilaniu et al., 2003*; *Fuentealba et al., 2008*; *Rujano et al., 2006*; *Singhvi and Garriga, 2009*; *Zhou et al., 2014*).

The ER processes one-third of cellular proteome and possesses stronger ability to maintain aggregation-prone proteins in a non-toxic state than the cytosol owing to its specific molecular chaperone environment (*Vincenz-Donnelly et al., 2018*). Chaperones in the ER such as HSPA5/BiP regulates the unfolded protein response (UPR), which blocks instant protein synthesis and activates transcription of genes involved in PQC (*Preissler and Ron, 2019*; *Wiseman et al., 2022*). In addition, the proteostatic stress in the ER triggers ER-associated degradation (ERAD) and ER-phagy to remove misfolded proteins (*Mochida and Nakatogawa, 2022*; *Olzmann et al., 2013*). Despite the high capacity of proteostatic stress response in the ER (*Rousseau et al., 2004*; *Vincenz-Donnelly et al., 2018*), excessive misfolded proteins still lead to formation of protein aggregates in the ER (*Melo et al., 2022*; *Miyata et al., 2020*). In response to protein aggregation in the ER, BiP functions as a disaggregase to promote solubilization of these aggregates. In cells undergoing asymmetric cell division, such as in the budding yeast, the ER diffusion barrier and the ER stress surveillance (ERSU) pathway ensure ER protein aggregates are retained preferentially in the short-lived mother cells during cell division (*Clay et al., 2014*; *Piña and Niwa, 2015*). How ER protein aggregates are regulated in dividing cells, especially those that do not undergo asymmetric cell division, is less well studied. We addressed the fate of ER protein aggregates in human cells undergoing cell division symmetrically and identified a clearance mechanism of protein aggregates when cells are progressing through mitosis and cytokinesis.

## Results

### Targeting a proteostasis reporter to the ER results in protein aggregate formation in the nucleus

To investigate how cells maintain their proteostasis during cell division, we employed a proteostasis reporter consists of a firefly luciferase mutant prone for protein misfolding and protein aggregation fused to a green fluorescent protein (FlucDM-eGFP) (*Gupta et al., 2011*). Next, we targeted the reporter to the ER by fusing an ER-targeting sequence at its N-terminus (ER-FlucDM-eGFP). Interestingly, we found that ER-FlucDM-eGFP assembled into visible protein aggregates in the nucleus when stably expressed in mammary epithelial MCF10A cells in the absence of any perturbation of the proteome stability (*Figure 1A*). The protein aggregates were below detection limit of our microscopy in the first 2 d after lentivirus transduction but progressively accumulated in cells and stabilized in the population several days after lentivirus transduction. This contrasted with a previous study in which the ER-FlucDM-eGFP was transiently expressed in cells and hence did not allow long-term tracking of the protein (*Sharma et al., 2018*). ER-FlucDM-mCherry localized to the ER network indicated by the ER membrane protein Sec61β (*Figure 1B*). Since we detected aggregate formation in the nucleus in cells expressing ER-FlucDM-eGFP, we tested if targeting FlucDM-eGFP to the nucleus (NLS-FlucDM-eGFP) would result in the formation of visible protein aggregates. Our data showed that NLS-FlucDM-eGFP did not form protein aggregates in the nucleus and was not co-localized with the ER-FlucDM-mCherry and accumulated lower protein abundance than ER-FlucDM-eGFP (*Figure 1— figure supplement 1A–D*).

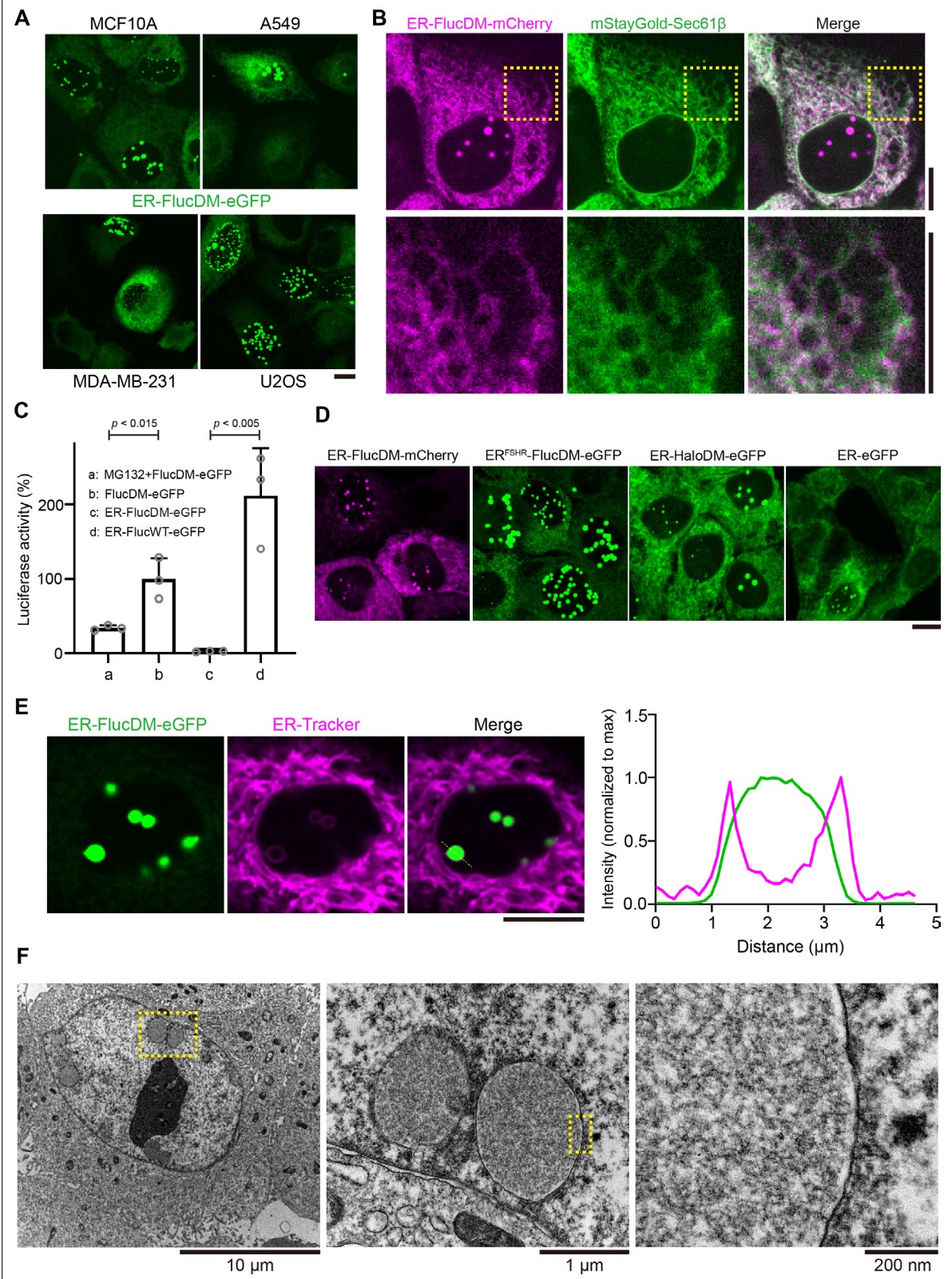

**Figure 1.** ER-FlucDM-eGFP forms protein aggregates in the nucleus. (**A**) Representative images of ER-FlucDM-eGFP expressed in different cell lines. Maximum projected images were shown. (**B**) Microscopy images of cells expressing ER-FlucDM-mCherry and mStayGold-Sec61β. Single Z-slice images were shown. (**C**) Quantification of the luciferase activity in MCF10A cells expressing different constructs. Unpaired t-test. Data were collected from three independent experiments. Abundance of GFP fusion proteins was determined using protein blotting and was used in normalization. (**D**) Representative

*Figure 1 continued on next page*

*Figure 1 continued*

images to show MCF10A cells expressing ER-FlucDM-mCherry, ER[FSHR]-FlucDM-eGFP (signal peptide sequence from the N-terminus of FSHR), ER-HaloDM-eGFP or ER-eGFP. Maximum projected images were shown. (**E**) Images of ER-FlucDM-eGFP aggregates surrounded by the endoplasmic reticulum (ER) membrane in a MCF10A cell. Single Z-slice images were shown. (**F**) Transmission electron microscopy (TEM) images of sectioned MCF10A interphase cells expressing ER-FlucDM-eGFP. Data represent mean + SD. Scale bar, 10 μm (**A to E**).

The online version of this article includes the following source data and figure supplement(s) for figure 1:

**Source data 1.** PDF file containing original western blots corresponding to *Figure 1C*.

**Source data 2.** Original files for western blot analysis displayed in *Figure 1C*.

**Figure supplement 1.** ER-FlucDM-eGFP forms protein aggregates, related to *Figure 1* (**A**) Microscopy images of cells expressing NLS-FlucDM-eGFP.

**Figure supplement 1—source data 1.** PDF file containing original membranes corresponding to *Figure 1—figure supplement 1C and G*.

**Figure supplement 1—source data 2.** Original files for western blot analysis displayed in *Figure 1—figure supplement 1C and G*.

Next, we expressed pathological mutants of cystic fibrosis transmembrane conductance regulator (CFTR) and Alpha-1 antitrypsin (AAT), which are CFTR-ΔF508 and AAT S or Z variants, respectively. These pathological mutants misfold and accumulate in the ER (*Greene et al., 2016*; *Lukacs and Verkman, 2012*). However, expression of CFTR-Δ508 and AAT S or Z variants did not result in the aggregate accumulation in the nucleus as observed in ER-FlucDM-eGFP (*Figure 1—figure supplement 1E and F*). Thus, a pathological mutant of ER proteins that are confined in the nucleus remains to be determined. Luciferase activity measurement showed that ER-FlucDM-eGFP has significantly reduced enzymatic activity and was not functional possibly due to misfolding and protein aggregation (*Figure 1C*). ER-FlucDM-eGFP was also detected in the insoluble fraction after heat stress (*Figure 1—figure supplement 1G*). Moreover, we observed the colocalization of ER-FlucDM-eGFP aggregates and thioflavin T (ThT), a universal dye to detect amyloid fibrils, further supporting these aggregates were misfolded protein aggregates (*Figure 1—figure supplement 1H*). Thus, expression of ER-FlucDM-eGFP results in proteostatic stress and allows us to study how protein aggregates are regulated in human cells.

Formation of protein aggregates in the nucleus by ER-FlucDM-eGFP was not limited to MCF10A cells. The protein aggregates were also detected in the nucleus when ER-FlucDM-eGFP was expressed in other cell lines such as A549, MDA-MB-231, and U2OS cells (*Figure 1A*, *Figure 1—figure supplement 1I*). Similar observations were found when a different N-terminal signal peptide sequence was used (*Figure 1D*, *Figure 1—figure supplement 1J*). In addition, when ER-FlucDM was replaced by ER-HaloDM, which is a modified haloalkane dehalogenase prone to misfolding and aggregation (*Melo et al., 2022*), we observed similar protein aggregates in the nucleus (*Figure 1D*, *Figure 1—figure supplement 1K*). However, only a very low number of cells formed aggregates in the nucleus when intact GFP was targeted to the ER (*Figure 1—figure supplement 1K*). Thus, ER-FlucDM-eGFP formed protein aggregates in the nucleus independent of cell line, fluorescent protein types, specific ER targeting sequence, and FlucDM.

Despite localizing in the nucleus, we found that the protein aggregates were surrounded by the ER membrane as stained by the ER-tracker (*Figure 1E*). To further confirm this, we sectioned the cells and used electron microscopy to observe the protein aggregates. We found that the aggregates were surrounded by a single-layer membrane closed to the inner membrane of the nucleus (*Figure 1F*). Thus, the aggregates are an intra-nuclear membranous structure surrounded by the ER membrane.

## Differential and slow turnover of protein aggregates in cells at interphase and mitosis

Pathological proteins aggregates such as Z-α1-antitrypsin accumulated in the ER exhibited minimal turnover (*Dickens et al., 2016*). To test whether ER-FlucDM-eGFP aggregates have reduced dynamics as well, we performed fluorescence recovery after photobleaching (FRAP) in cells expressing both ER-FlucDM-eGFP and ER-FlucDM-mCherry. The ER-FlucDM-eGFP signal served as a tracker for the location of aggregates and the ER-FlucDM-mCherry was used to probe the aggregate dynamics. We photobleached ER-FlucDM-mCherry with 561 nm laser while ensuring ER-FlucDM-eGFP signal was not affected. In interphase cells, ER-FlucDM-mCherry aggregates intensity barely recovered after 4.5 min post-bleaching, indicating that ER-FlucDM-mCherry aggregates possessed minimal turnover similar to that of other protein aggregates such as Z-α1-antitrypsin (*Figure 2A*). Similarly, ER-HaloDM-eGFP

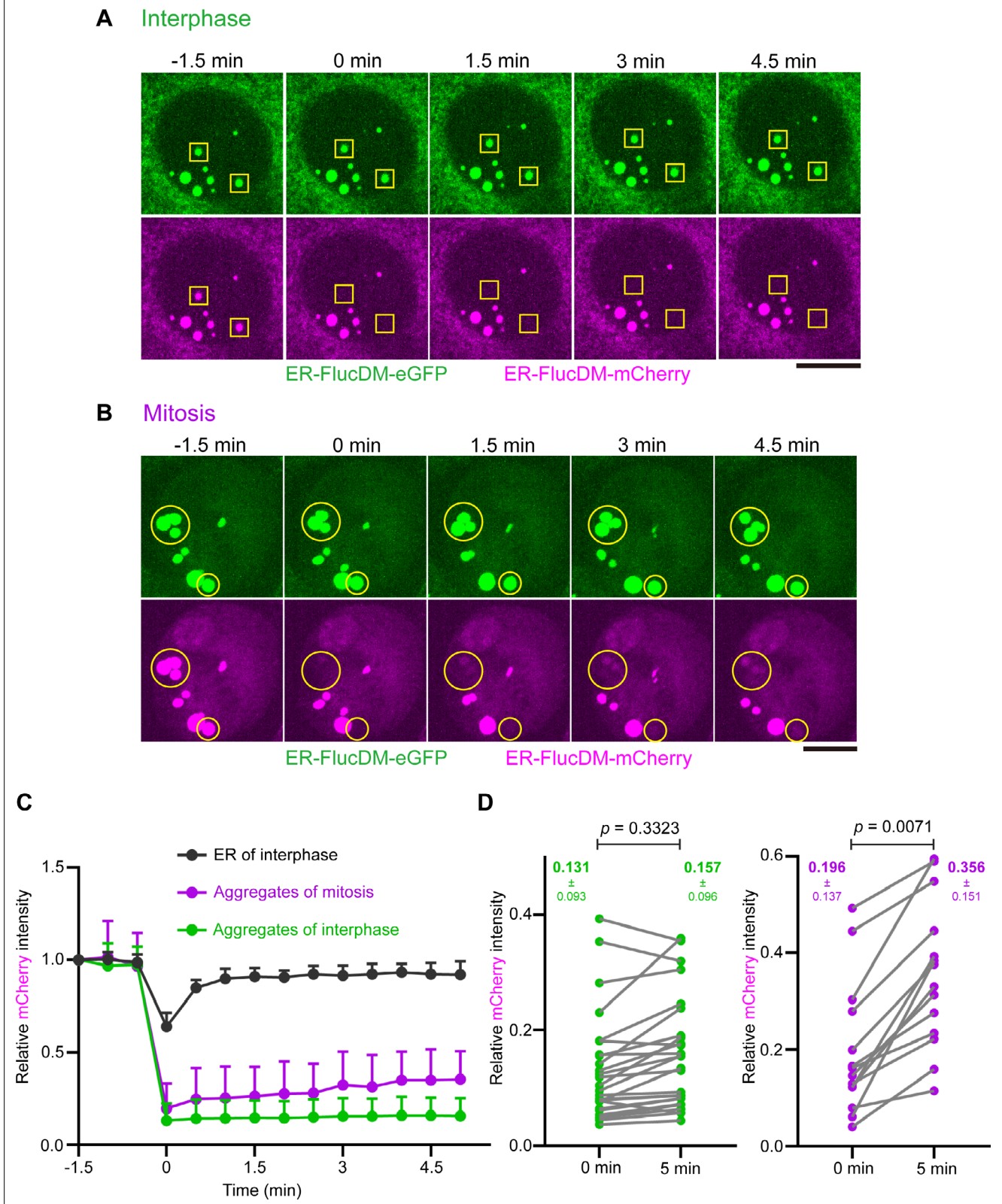

**Figure 2.** Differential aggregate dynamics in cells at interphase and mitosis. (**A**) Representative images of interphase cells co-expressing ER-FlucDM-eGFP and ER-FlucDM-mCherry before and after photobleaching. Yellow rectangles indicate bleached aggregates. Maximum projected images were shown. (**B**) Representative images of mitotic cells co-expressing ER-FlucDM-eGFP and ER-FlucDM-mCherry before and after photobleaching. Yellow circles indicate bleached aggregates. Maximum projected images were shown. (**C**) Recovery curves of ER-FlucDM-mCherry intensity after

*Figure 2 continued on next page*

*Figure 2 continued*

photobleaching. 14 region of interests (ROIs) from the endoplasmic reticulum (ER) of interphase cells, 14 aggregates from mitotic cells, 26 aggregates from interphase cells. Data represent mean + SD. A small region of ER-FlucDM-mCherry in the ER network was chosen as a control for photobleaching. (**D**) Quantification of the ER-FlucDM-mCherry intensity of bleached aggregates right after bleaching and after 5 min of recovery. 0 min and 5 min refer to the time points in (**C**). Paired t-test (two-tailed). Data were collected from four independent experiments. Scale bar, 10 µm.

The online version of this article includes the following figure supplement(s) for figure 2:

**Figure supplement 1.** Fluorescence recovery after photobleaching (FRAP) analysis of protein aggregates, related to *Figure 2*.

aggregates in the nucleus also displayed low recovery after photobleaching (*Figure 2—figure supplement 1A*). When ER-FlucDM-mCherry aggregates were photobleached in mitotic cells, the recovery of intensity was also much lesser than the ER-FlucDM-mCherry signal present in the ER network (*Figure 2B and C*). However, ER-FlucDM-mCherry aggregates in mitotic cells showed a higher recovery than that of in interphase cells (*Figure 2D*; aggregates of interphase cells recovered from 0.131±0.093–0.157±0.095; aggregates of mitotic cells recovered from 0.196±0.137–0.356±0.151).

Consistently, the single-cell analysis revealed that in mitotic cells, there was an increase in the recovery intensity of ER-FlucDM-mCherry in late time points as compared to that in early time points after photobleaching. The increased recovery intensity was not observed in ER-FlucDM-mCherry aggregates of interphase cells (*Figure 2—figure supplement 1B–D*). Taken together, ER-FlucDM-mCherry aggregates in the nucleus have low turnover and the aggregate displays increased turnover in mitotic cells.

## Protein aggregates are cleared during cell division

Since aggregates displayed some levels of turnover during mitosis (*Figure 2B–D*), we next investigated the aggregate behavior throughout cell division. To this end, we labeled cells expressing ER-FlucDM-eGFP with SiR-Tubulin to distinguish cells at various stages based on the microtubule organization. We categorized cells into interphase and prophase, prometaphase, metaphase, anaphase, telophase, and early G1 cells based on their microtubule structures (*Figure 3A*). Next, we analyzed the number and area of aggregates in cells of each category. Interestingly, we observed that the number and area of aggregates in telophase and early G1 cells were significantly lower than in interphase cells, suggesting that aggregates are cleared gradually when cells progress through mitosis and cytokinesis (*Figure 3B*).

To test whether this was the case, we employed time-lapse imaging to profile aggregates in cells undergoing division. To increase the number of dividing cells during imaging, we synchronized cells at G2/M and released them into mitosis prior to imaging. We observed that upon entry into mitosis, the ER-FlucDM-eGFP aggregates were released from the nucleus during nuclear envelope breakdown and gradually decreased in numbers as cells progress through mitosis and cytokinesis (*Figure 3C* and *Videos 1–3*). In majority of early divided G1 cells, there was essentially no detectable aggregates (*Figure 3D*). Interestingly, the aggregates released from the nucleus during mitosis were still surrounded by the ER membrane as indicated by the ER-tracker (*Figure 3—figure supplement 1A and B*). The decrease of aggregates appears to happen specifically in dividing cells as the number and area of aggregates remained largely unchanged in interphase cells (*Figure 3C and D*). Furthermore, aggregates formed in cells expressing ER-HaloDM-eGFP decreased in numbers when cells progress through mitosis and cytokinesis (*Figure 3—figure supplement 1C* and *Video 2*), suggesting that the clearance of protein aggregates during cell division was not specific to ER-FlucDM-eGFP.

## Effects of ER stress inducers on aggregate clearance in mitotic cells

Expression of ER-FlucDM-eGFP results in proteostatic stress in cells. Consistently, we observed that gene expression for chaperones and co-chaperones in the ER were upregulated in these cells (*Figure 4A*), indicating cells involved in ER proteostasis control during adaptation. Proteomic analysis of cells expressing ER-FlucDM-GFP showed up-regulation of proteins involved in ER stress response (*Figure 4—figure supplement 1A*). We speculated that acute perturbation to ER homeostasis could exacerbate the proteostatic stress in dividing cells expressing ER-FlucDM-eGFP and affect aggregate clearance during mitosis. To test if this was the case, synchronized cells expressing ER-FlucDM-eGFP were released into mitosis in the presence of either DMSO (control) or 1 µM thapsigargin (Thaps), which blocks the ER calcium ion pump and causes ER stress. In cells at prometaphase and metaphase

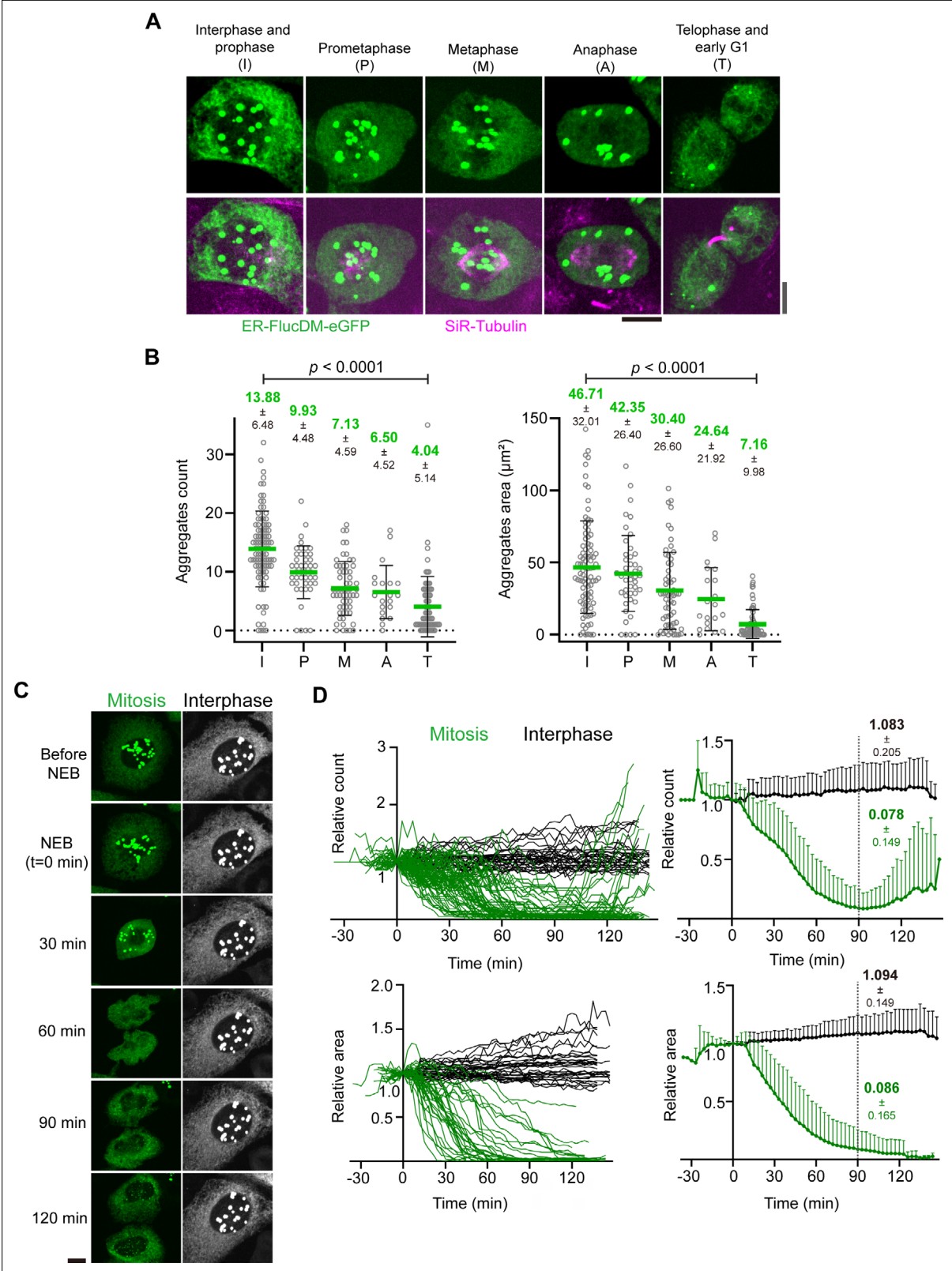

**Figure 3.** Aggregates are cleared during cell division. (**A**) Representative images of fixed dividing ER-FlucDM-eGFP cells at different cell cycle phases. Gray scale bar: telophase and early G1 cells; black scale bar: interphase and prophase, prometaphase, metaphase, anaphase. Maximum projected images were shown. Scale bar, 10 μm. (**B**) Quantification of ER-FlucDM-eGFP aggregates from dividing cells. Count and area are individual cell data. Each gray ring represents aggregates from one cell. Interphase and prophase, 91 cells; prometaphase, 45 cells; metaphase, 60 cells; anaphase,

*Figure 3 continued on next page*

*Figure 3 continued*

20 cells; telophase and early G1, 73 cells. From three independent experiments. Unpaired two-side Mann-Whitney U-test. Data represent mean ± SD (**C**) Representative time-lapse images of cells expressing ER-FlucDM-eGFP released from the G2/M boundary. NEB (nuclear envelope breakdown) is set to be t=0 min. Maximum projected images were shown. Scale bar, 10 µm. (**D**) Quantification of aggregate number and area of dividing cells expressing ER-FlucDM-eGFP. Left panel: data of individual cells. Right panel: mean values + SD of single-cell data (left panel). Time point of NEB in each cell was set to 0 min. Aggregate count and total area of each cell at each time point is normalized to its own value at t=0 min. Aggregate count data are from 61 interphase and 73 dividing cells of four independent experiments. Aggregate area of each cell is calculated from 32 interphase and 34 dividing cells of two independent experiments.

The online version of this article includes the following figure supplement(s) for figure 3:

**Figure supplement 1.** Aggregates in dividing cells, related to *Figure 3*.

(35 min after release), DMSO and Thaps treatments showed comparable number and area of aggregates in cells (*Figure 4B and C*). However, in cells at the late stage of cell division (65 min after release), there were significantly higher number of aggregates retained in the cytosol of Thaps-treated cells than in DMSO (*Figure 4B and C*). Consistently, time-lapse microscopy revealed that ER-FlucDM-eGFP aggregates retained in the cytosol after cytokinesis in Thaps-treated cells, whereas the aggregates were largely cleared in control cells completing cytokinesis (*Figure 4D and E*, *Figure 4—figure supplement 1B* and *Video 4*). Thus, acute treatment of ER stress inducer Thaps in dividing cells prevents clearance of ER-FlucDM-eGFP aggregates.

A previous study showed that cells with protein aggregates treated for hours with Thaps or Tunicamycin (Tuni), which inhibits protein glycosylation in the ER, promotes aggregate clearance in cells (*Melo et al., 2022*). To test if prolonged treatment of cells expressing ER-FlucDM-eGFP with Tuni could promote aggregate clearance, we pretreated cells blocked at G2/M with varying concentrations of Tuni for 3 hr and released cells into mitosis. Interestingly, prolonged pretreatment of Tuni prior to mitosis promoted aggregate clearance in dividing cells (*Figure 4—figure supplement 1C*). Thus, acute treatment of cells expressing ER-FlucDM-eGFP with ER stress inducers prevents aggregate clearance in dividing cells while prolonged treatment with ER stress inducers promotes aggregate clearance.

## Hsp70 family protein BiP is required for aggregate clearance during cell division

Since we observed that the expression of a chaperone Hsp70 (HSPA5/BiP) was upregulated in cells expressing ER-FlucDM-eGFP (*Figure 4A*) and Hsp70 is involved in elimination of protein aggregates (*Melo et al., 2022*; *Nillegoda et al., 2015*), we next tested whether the clearance of ER-FlucDM-eGFP aggregates in dividing cells was mediated by Hsp70 family proteins. To this end, cells entering

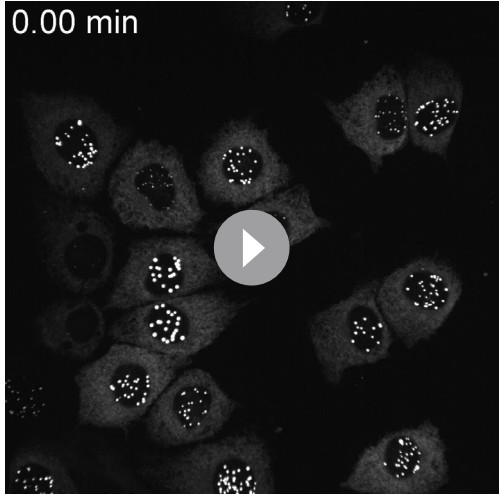

**Video 1.** ER-FlucDM-eGFP in cells undergoing mitosis and cytokinesis.

https://elifesciences.org/articles/96675/figures#video1

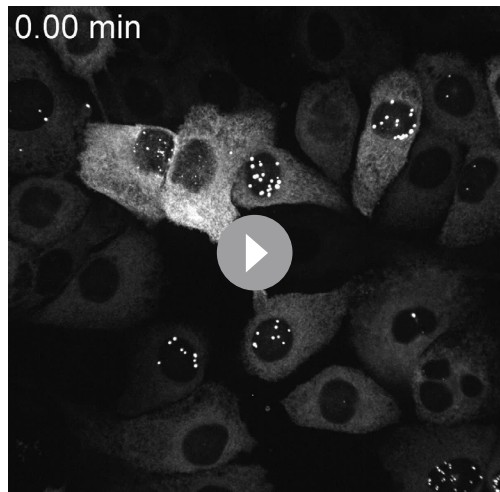

**Video 2.** ER-HaloDM-eGFP in cells undergoing mitosis and cytokinesis.

https://elifesciences.org/articles/96675/figures#video2

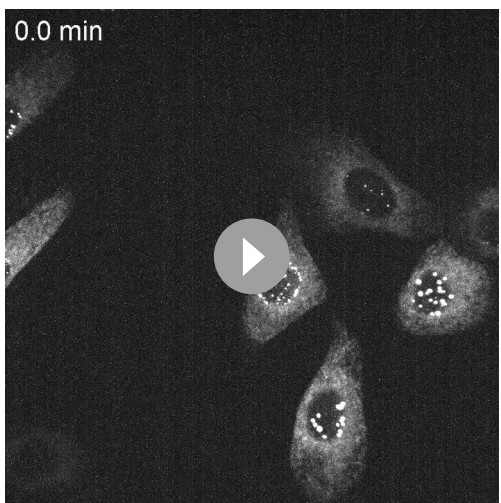

**Video 3.** ER-FlucDM-eGFP in cells without synchronization.
https://elifesciences.org/articles/96675/figures#video3

mitosis were treated with either DMSO (control) or VER-155008 (VER), an inhibitor of HSP70's ATPase activity (*Samanta et al., 2021*). We found that VER-treated dividing cells fixed after 65 min released from the G2/M boundary retained a high number of ER-FlucDM-eGFP aggregates in the cytosol compared to control treatment (*Figure 5A and B*). Time-lapse microscopy showed that when cells were treated with VER, ER-FlucDM-eGFP aggregates released from the nucleus upon entry into mitosis were not cleared to the same extent as control cells (*Figure 5C* and *Video 5*).

HSPA5/BiP is a major Hsp70 family member in regulating ER proteostasis. We next tested whether inhibition of BiP function could affect the clearance of aggregates in dividing cells. Similar to VER treatment, inhibition of BiP's ATPase activity by YUM-70 (*Samanta et al., 2021*) led to a dose-dependent accumulation of ER-FlucDM-eGFP aggregates in cells (*Figure 5D and E*). Moreover, by using immunofluorescence staining, we found a strong colocalization of BiP and

ER-FlucDM-eGFP, consistent with a role of BiP in regulating ER-FlucDM-eGFP aggregates (*Figure 5—figure supplement 1A–C*). Collectively, our data demonstrated that the clearance of ER-FlucDM-eGFP aggregates was regulated by the HSP70 family proteins, particularly BiP.

## Aggregate clearance is a mitotic exit event but is independent of the APC/C

Protein aggregates are subjected to proteasomal degradation in cells (*Tyedmers et al., 2010*). To test if aggregate clearance during cell division involves proteasomes, we inhibited the proteasome activity with MG132 in dividing cells expressing ER-FlucDM-eGFP. After treating cells released from the G2/M boundary with low or high concentrations of MG132 for 55 min, majority of mitotic cells were arrested at metaphase and accumulated a high number of aggregates as compared to control cells treated with DMSO (*Figure 6—figure supplement 1A and B*). Next, we inhibited the proteasome with MG132 in cells at late anaphase or telophase and examined the aggregates 35 min after MG132 treatment (*Figure 6—figure supplement 1C–F*). In cells completing cytokinesis, there were significantly high number of aggregates in the early divided cells treated with MG132 versus the control (*Figure 6E*, *Figure 6—figure supplement 1D, G and H*).

Since protein aggregates are cleared when cells progress through mitosis and cytokinesis, which are the cell cycle stages with low Cyclin B/Cdk1 activity, we speculated that inactivation of Cdk1 activity in MG132-treated and metaphase-arrested cells could lead to clearance of ER-FlucDM-eGFP aggregates in the cytosol. To test if this was the case, we released cells into mitosis in the presence of MG132 for 55 min and treated metaphase-arrested cells with RO-3306 to inhibit Cdk1 activity (*Figure 6A*). Interestingly, ER-FlucDM-eGFP aggregates were cleared in cells treated with RO-3306 while the aggregates were retained in cells treated with DMSO (*Figure 6B*, *Figure 6—figure supplement 1I*). Consistently, live cell imaging demonstrated similar accelerated clearance in Cdk1-inhibited cells (*Figure 6C and D*). Thus, clearance of the aggregates happens when cells exit from mitosis.

We established that clearance of aggregates during cell division requires proteasomes and Cdk1 inactivation, it is possible that the APC/C, which is an E3 ubiquitin ligase regulating mitotic exit (*Watson et al., 2019*), is involved in clearing the aggregates. To test if this was the case, we inhibited APC/C activity using a cocktail of APC/C inhibitors (APC/Ci) consists of Apcin and ProTAME in dividing cells expressing ER-FlucDM-eGFP (*Sackton et al., 2014*). Interestingly, inhibition of APC/C did not affect aggregate clearance as APC/Ci-treated cells that have arrested at metaphase have a comparable number of aggregates as in control DMSO-treated cells (*Figure 6E and F*). When cells at late anaphase or telophase were treated with APC/Ci, the newly divided cells had a similar number

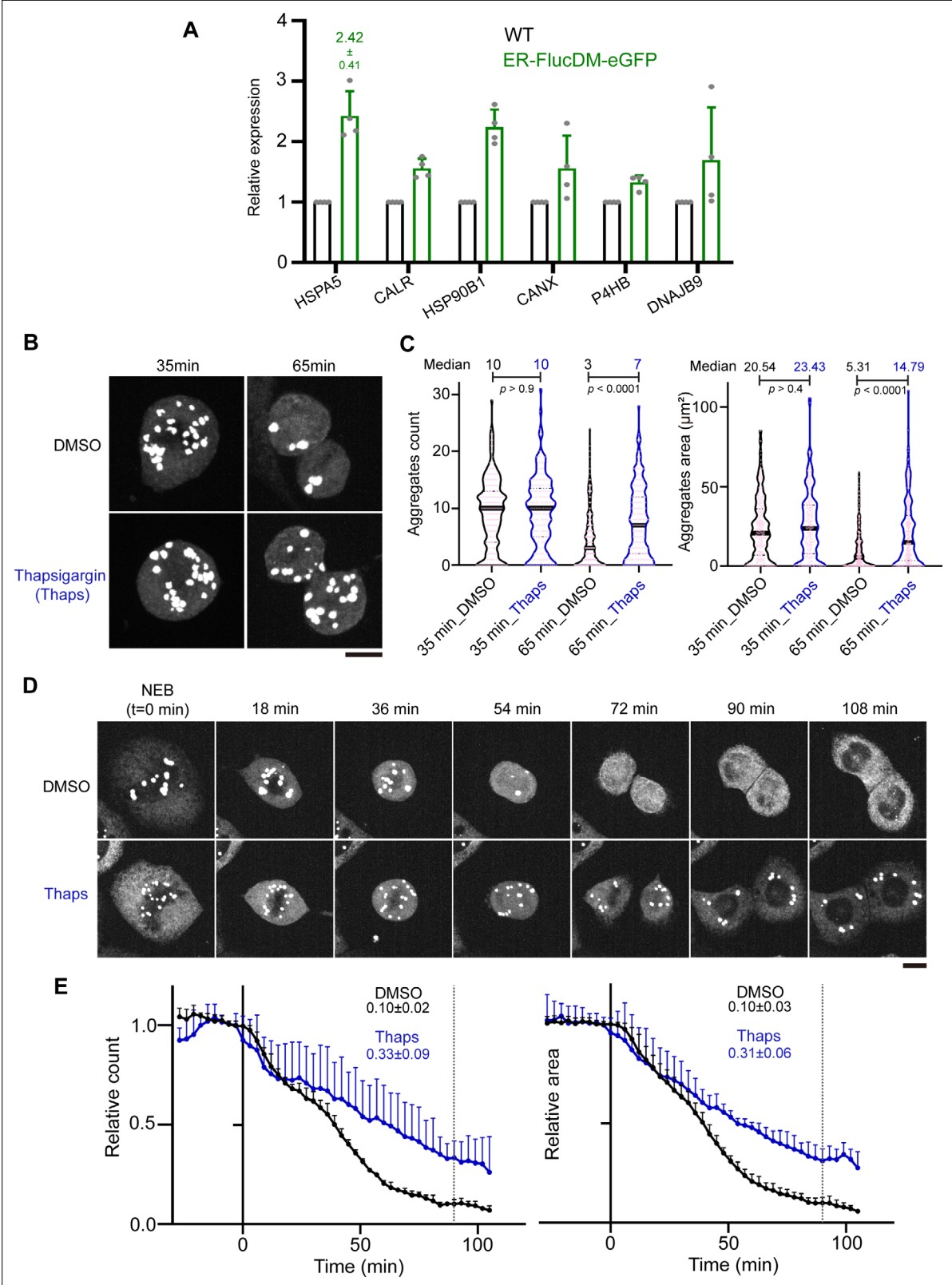

**Figure 4.** Effects of endoplasmic reticulum (ER) stressors on aggregate clearance. (**A**) Relative gene expression levels of chaperones and co-chaperones in cells expressing ER-FlucDM-eGFP. Four independent experiments. (**B**) Images of cells treated with DMSO or 1 μM Thaps and fixed at 35 min or 65 min after released from the G2/M boundary. Maximum projected images were shown. (**C**) Quantification of cells treated with DMSO or 1 μM Thaps and fixed at 35 min or 65 min after released from the G2/M boundary. 35 min_DMSO, 207 cells; 35 min_Thaps, 173 cells; 65 min_DMSO, 244 cells;

*Figure 4 continued on next page*

*Figure 4 continued*

65 min_Thaps, 209 cells. Unpaired two-side Mann-Whitney U-test. The solid black line in violin plot indicates median of data in the column. Gray dotted lines are quartiles. Each light pink circle indicates measurements of individual cells. (**D**) Representative time-lapse images of cells expressing ER-FlucDM-eGFP treated with DMSO or 1 μM Thaps after released from the G2/M boundary. Maximum projected images were shown. (**E**) Quantification of cells expressing ER-FlucDM-eGFP treated with DMSO or 1 μM Thaps. Time point of nuclear envelope breakdown (NEB) in each cell was set to 0 min. Aggregates count and total area of each cell is normalized to its own value at t=0 min. Black dotted lines are 90 min since NEB. DMSO, 89 cells; Thaps, 65 cells. Scale bar, 10 μm. Three (**C** and **E**) or four (**A**) independent experiments.

The online version of this article includes the following figure supplement(s) for figure 4:

**Figure supplement 1.** Effect of Thapsigargin or Tunicamycin on aggregates clearance, related to *Figure 4*.

---

of aggregates as in control cells (*Figure 6G–I*). Lastly, we verified the efficacy of APC/C inhibition using cells expressing the APC/C reporter derived from Geminin (*Figure 6—figure supplement 1J*). Consistently, APC/Ci treatment has prevented the removal of APC/C reporter in newly divided cells indicating that the APC/C was inhibited by the dosage of APC/Ci used in our experiments (*Figure 6— figure supplement 1K*). Taken together, our data showed that clearance of ER-FlucDM-eGFP aggregates happens when cells exit from mitosis, is proteasome-dependent but does not involve APC/C.

To examine if ER-FlucDM-eGFP decreased in abundance while cells progressing through mitosis, we inhibited protein translation using cycloheximide (CHX) in cells released from G2/M boundary. Short half-life proteins like p53 showed a rapid decrease of abundance upon CHX treatment, indicating CHX prevented new protein synthesis in our treatment (*Figure 6—figure supplement 2A and B*). ER-FlucDM-eGFP did not show a significant change in its abundance after CHX treatment, suggesting that ER-FlucDM-eGFP aggregates were not cleared by the protein degradation mechanism directly (*Figure 6—figure supplement 2A and C*). Consistently, when the total GFP intensity of ER-FlucDM-eGFP was quantitated in dividing cells 20–30 min (early) or 85–95 min (late) released from G2/M boundary, there was no significant change in GFP fluorescence intensity between early and late dividing cells (*Figure 6—figure supplement 2D*), further supported that the protein degradation mechanism was not responsible for the ER-FlucDM-eGFP aggregate clearance.

## Aggregate clearance depends on the ER reorganization during cell division

Since we showed that ER-FlucDM-eGFP aggregates are not primarily cleared by the protein degradation mechanism, we considered the possibility that aggregate clearance requires reorganization of the ER that happens in cells exiting mitosis. To this end, we used the CRISPR interference (CRISPRi) technique to knock down ER membrane fusion proteins Atlastin 2 (ATL2) and Atlastin 3 (ATL3) in cells expressing ER-FlucDM-eGFP and quantitated the aggregate numbers in cells exiting mitosis (65 min after release). Interestingly, dividing cells depleted of ATL2 and ATL3 accumulated more ER-FlucDM-eGFP aggregates than in control cells (*Figure 7A and B*).

The mitotic kinase Aurora A has been recently shown to remodel the ER network during mitosis (*Zhang et al., 2024*). We next tested if blocking Aurora A activity could affect aggregate clearance. When mitotic cells expressing ER-FlucDM-eGFP were treated with Aurora A inhibitor, MLN-8237, aggregate clearance was prevented compared to cells treated with DMSO (*Figure 7C and D*). Since the ER reorganization involves microtubule dynamics, we also perturbed microtubule networks by treating cells expressing ER-FlucDM-eGFP with nocodazole (*Figure 7E and F*). Our data showed that mitotic cells treated with nocodazole accumulated a higher number of

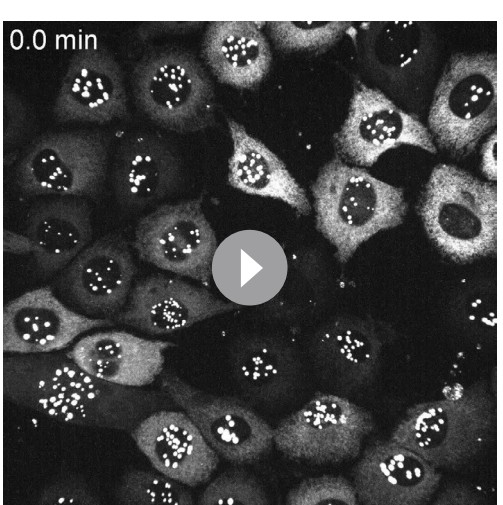

**Video 4.** ER-FlucDM-eGFP aggregates in Thaps-treated cells.

https://elifesciences.org/articles/96675/figures#video4

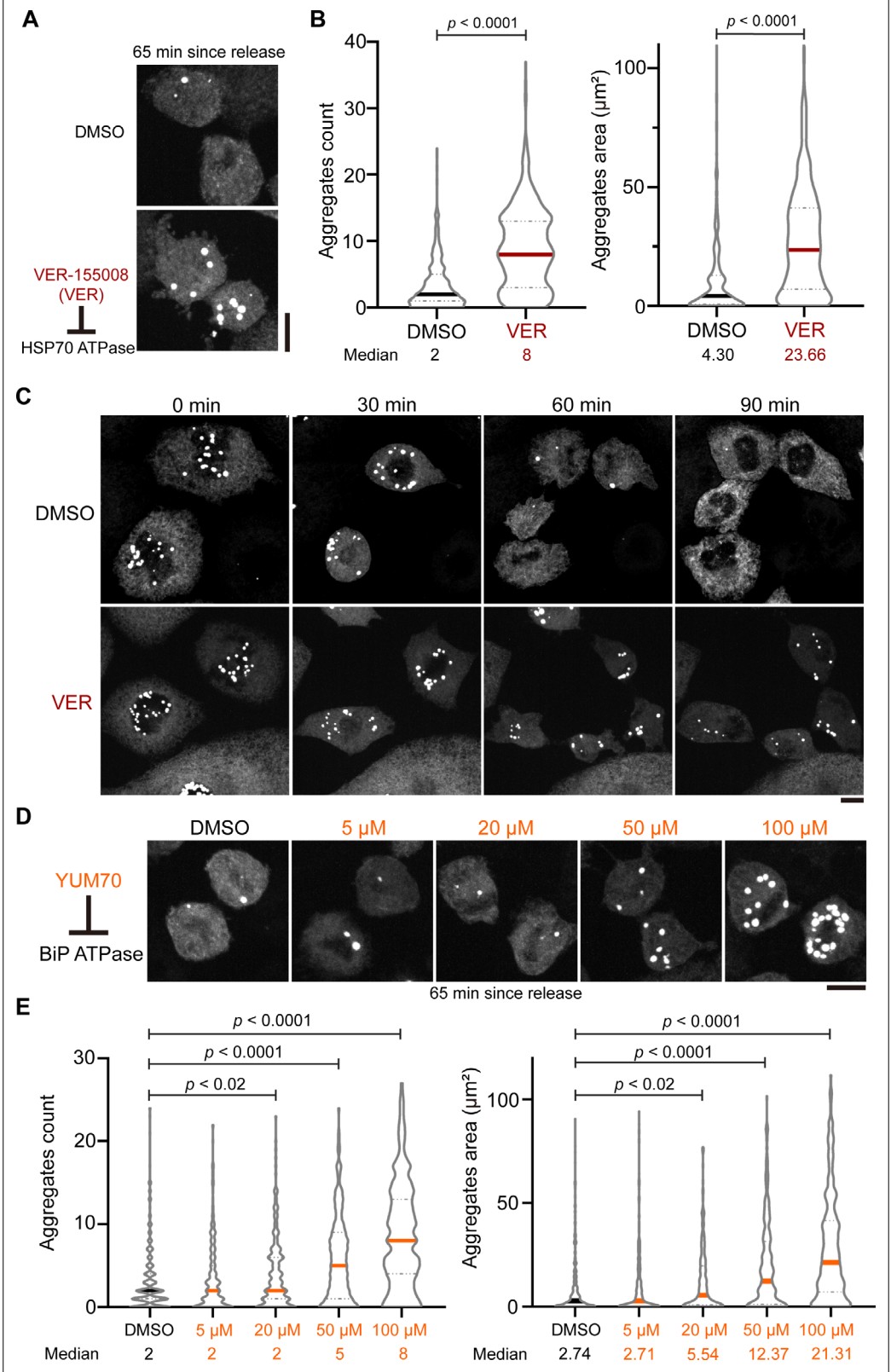

**Figure 5.** Inhibition of HSP70 and BiP prevents aggregate clearance. (**A**) Representative images of cells expressing ER-FlucDM-eGFP treated with DMSO or 50 μM VER and fixed at 65 min after released from the G2/M boundary. Maximum projected images were shown. (**B**) Quantification of cells expressing ER-FlucDM-eGFP treated with DMSO or 50 μM VER and fixed at 65 min after released from the G2/M boundary. DMSO, 357 cells; VER, 465 cells.

*Figure 5 continued on next page*

*Figure 5 continued*

(**C**) Time-lapse images of dividing cells expressing ER-FlucDM-eGFP treated with DMSO or 50 µM VER. Maximum projected images were shown. (**D**) Representative images of cells expressing ER-FlucDM-eGFP treated with DMSO or different concentrations of YUM70. Maximum projected images were shown. (**E**) Quantification of cells expressing ER-FlucDM-eGFP treated with DMSO or YUM70 and fixed at 65 min after released from the G2/M boundary. DMSO, 310 cells; 5 µM YUM70, 258 cells; 20 µM YUM70, 215 cells; 50 µM YUM70, 209 cells; 100 µM YUM70, 287 cells. Scale bar, 10 µm. Unpaired two-side Mann-Whitney U-test. Each solid line in violin plot indicates median of data in the column. Gray dotted lines are quartiles. (**A, B, D, and E**) are from three independent experiments. (**C**) is from two independent experiments.

The online version of this article includes the following figure supplement(s) for figure 5:

**Figure supplement 1.** Colocalization of BiP and ER-FlucDM-eGFP, related to *Figure 5*.

---

aggregates. Taken together, we found that clearance of ER-FlucDM-eGFP aggregates involves reorganization of the ER during mitotic exit (*Lee et al., 1989*).

## Discussion

We showed that stable expression of ER-FlucDM-eGFP leads to proteostatic stress and formation of protein aggregates in human cells without the use of additional proteostatic stressors. ER-FlucDM-eGFP aggregates may derive from the invagination of the inner nuclear membrane. *Morris et al., 1997* reported a similar aggregate formation in the nucleus upon BiP overexpression. This fluorescent-based live-cell imaging reporter enables us to study how protein aggregates are regulated in dividing cells, which are sensitive to genetic or environmental perturbations. Conventional methods that use proteasomal inhibitors to cause proteome imbalance are not ideal to investigate proteostasis in dividing cells given an essential role of proteasomes in driving mitotic exit (*Ghislain et al., 1993*; *Glotzer et al., 1991*). Additionally, methods such as heat shock also perturb mitotic progression, especially in human cells (*Kakihana et al., 2019*). By using ER-FlucDM-eGFP that does not perturb cell cycle progression significantly, we reveal an unexpected process of clearing protein aggregates in cells progressing through mitosis and cytokinesis. We found that ER-FlucDM-eGFP aggregates are confined in the nucleus within the ER membrane during interphase. When cells are progressing from mitosis to cytokinesis in which the Cdk1 activity is decreasing, the ER networks reorganize allowing BiP that has disaggregation ability to clear the ER-FlucDM-eGFP aggregates.

Previous studies showed that the proteome of mitotic cells has higher structural stability and is less aggregation-prone compared to that of in interphase cells (*Becher et al., 2018*; *Wirth et al., 2013*). Post-translational modifications such as phosphorylation of mitotic proteins are suggested to be responsible for the protein stability during cell division (*Becher et al., 2018*). The aggregate clearance we identified in this study could represent an active mechanism that increases proteome stability in dividing cells. We found that aggregate clearance requires Cdk1 inactivation, which is a hallmark of mitotic exit, however, does not involve APC/C that is important in driving mitotic exit through protein degradation. It is possible that APC/C targets mostly cytosolic proteins whereas ER-FlucDM-eGFP aggregates are confined in a membrane during mitosis. Furthermore, the clearance of protein aggregates coincides with ER reorganization that happens throughout mitosis and cytokinesis, which may contribute to the aggregate clearance (*Bergman et al., 2015*; *Schlaitz et al., 2013*). A previous study showed that the size of protein aggregates is limited by the ER tubules (*Parashar et al., 2021*). In fact, when the ER reorganization

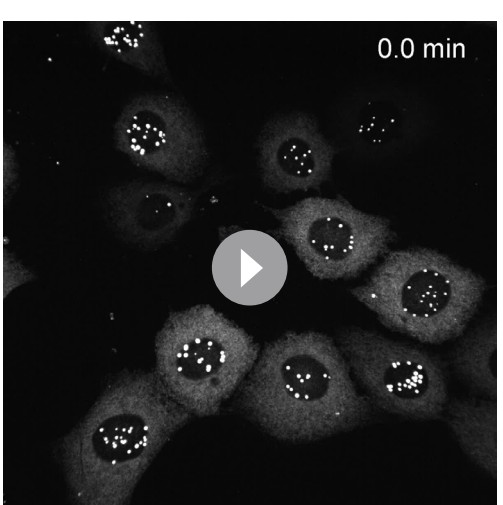

**Video 5.** ER-FlucDM-eGFP aggregates in VER-treated cells.

https://elifesciences.org/articles/96675/figures#video5

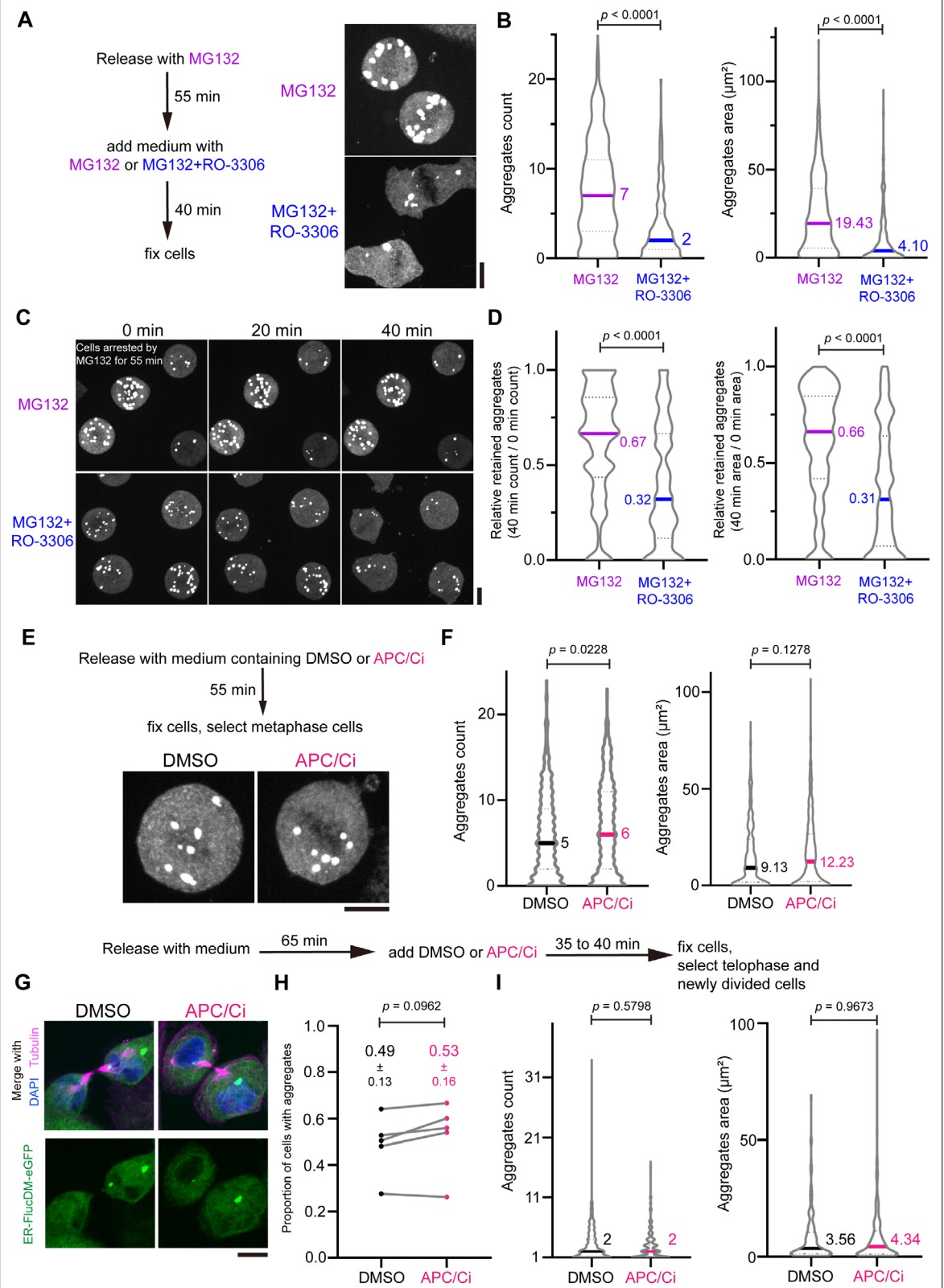

**Figure 6.** Effects of MG132, APCi, and RO-3306 (Cdk1i) on aggregate clearance. (**A**) Representative images of cells expressing ER-FlucDM-eGFP arrested at mitosis by MG132 and further treated with Cdk1i to induce mitotic exit. Cells were first treated with 25 µM MG132 for 55 min and mitotic arrested cells were further treated with 25 µM MG132 plus 10 µM RO-3306 or 25 µM MG132 only. Cells were fixed 40 min after the second treatment. Maximum projected images were shown. (**B**) Quantification of (**A**). MG132 only, 442 cells; MG132 +RO-3306, 374 cells. (**C**) Time-lapse images of cells

*Figure 6 continued on next page*

*Figure 6 continued*

expressing ER-FlucDM-eGFP arrested at mitosis by MG132 and further treated with RO-3306 to induce mitotic exit. Cells were treated with 25 μM MG132 for 55 min, and further treated with 25 μM MG132 plus 10 μM RO-3306 or 25 μM MG132 before imaging. Maximum projected images were shown. (**D**) Quantification of (**C**). Remaining aggregates at 40 min were calculated for their aggregates count or area and compared to that at 0 min. MG132 only, 125 cells; MG132 + RO-3306, 134 cells. (**E**) Images of cells expressing ER-FlucDM-eGFP treated with DMSO or APCi (20 μM proTAME and 300 μM Apcin) and fixed at 55 min after released from the G2/M boundary. Maximum projected images were shown. (**F**) Quantifications of cells expressing ER-FlucDM-eGFP treated with DMSO or APCi (20 μM proTAME and 300 μM Apcin). DMSO, 450 cells; APCi, 549 cells. (**G**) Images of cells at telophase or cells of newly divided upon treatment of DMSO or APCi for 35–40 min. Maximum projected images were shown. (**H**) Cells at telophase or newly divided cells that have aggregates upon treatment of DMSO or APCi were counted. (**I**) Aggregate number and area were counted in cells at telophase or in newly divided cells after treated with DMSO or APCi. Only cells with aggregates were included in the quantification. DMSO, 297 cells; APCi, 319 cells. Scale bar, 10 μm. Data are from at least three independent experiments and analyzed by unpaired two-side Mann-Whitney U-test (**B, D, F, and I**) or paired t-test (**H**).

The online version of this article includes the following source data and figure supplement(s) for figure 6:

**Figure supplement 1.** MG132, APCi, and RO-3306 treatments on aggregate clearance, related to *Figure 6*.

**Figure supplement 2.** Cycloheximide chase experiment and microscopy quantification of cells expressing ER-FlucDM-eGFP, related to *Figure 6*.

**Figure supplement 2—source data 1.** PDF file containing original membranes corresponding to *Figure 6—figure supplement 2A*.

**Figure supplement 2—source data 2.** Original files for western blot analysis displayed in *igure 6—figure supplement 2A*.

is perturbed by depleting ER membrane fusion proteins ATL2 and 3, clearance of ER-FlucDM-eGFP aggregates is affected. Consistently, ER-FlucDM-eGFP proteins do not turn over significantly in cells progressing through mitosis and cytokinesis. Thus, it is less likely that ER-FlucDM-eGFP is ubiquitinated by APC/C and targeted for protein degradation.

Misfolded or damaged proteins form protein aggregates and are spatially organized into specific subcellular sites to sequester them from other cellular processes (*Hill et al., 2017*). These aggregate deposition sites are localized in the cytosol or in the nucleus to confine misfolded proteins and aggregates for elimination from cells (*Kaganovich et al., 2008*). Similarly, we identified confinement of ER-FlucDM-eGFP aggregates in the nucleus and are surrounded by a single-layer membrane. These intranuclear membranous structures contain ER membranes and BiP, which is a key Hsp70 family chaperone in the ER and translocates into the nucleus under stress (*Liu et al., 2023*). How protein aggregates are targeted and assembled into the intranuclear membranous structure awaits future investigation. Interestingly, upon entry into mitosis, concomitant with the nuclear membrane breakdown, the membranous structure is released from the nucleus and the aggregates are cleared when cells progress through mitosis till early G1 phase of the daughter cells. The confinement and clearance of protein aggregates during cell division presumably protect dividing cells from harmful effects of protein misfolding and aggregation and ensure reliable inheritance of genetic materials. It has been reported that aggregates are asymmetrically retained in one of the daughter cells that are usually short-lived (*Rujano et al., 2006*). The aggregate clearance mechanism we report here may provide an additional cellular defense strategy to overcome proteostatic stress and to maintain proteome integrity during cell division. Proteomic analysis of cells expressing ER-FlucDM-eGFP identified up-regulation of multiple proteins involved in ER stress response, indicating that cells experience significant proteostatic stress upon expression of ER-FlucDM-eGFP. Moreover, several proteins regulating cell cycle displayed higher abundance in the proteome, which together with proteostatic stress indicate a perturbed cellular health. It is possible that higher proteostatic stress renders cells more sensitive to perturbation of the protein expression at the proteome level. Changes in cell cycle protein levels may affect the cell cycle transition and result in longer or shorter timeframes in certain cell cycle phases. Future studies addressing these will reveal the physiological consequences of aggregate clearance.

Hsp70 chaperones regulate proteostasis by preventing protein aggregation, promoting disaggregation, and subjecting solubilized aggregates to degradation or refolding (*Rosenzweig et al., 2019*). Our proteomic analysis revealed significant expression of BiP (HSPA5) in cells expressing ER-FlucDM-eGFP but not other Hsp70 family members. A recent study showed that BiP solubilizes aggregates and drives aggregate clearance in the ER (*Melo et al., 2022*). Our study of mitotic aggregate clearance in dividing cells further extend the role of BiP in disaggregation. Consistently, we found that Cdk1 inactivation in the presence of the proteasomal inhibitor could still lead to aggregate clearance, suggesting a partial role of the proteasome in clearing aggregates and the disaggregation breaks down the aggregates into smaller and possibly soluble species. The role of proteasome in

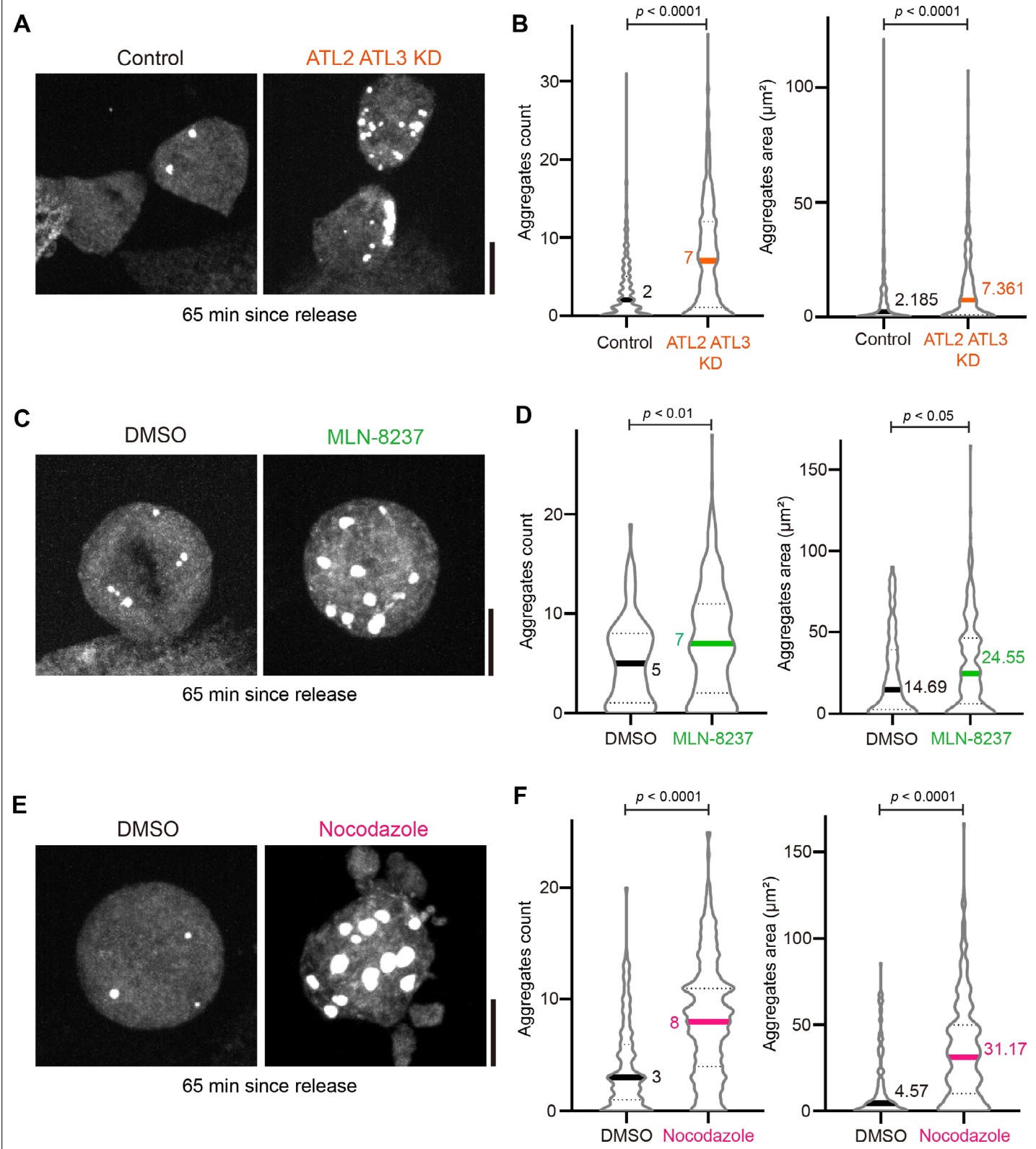

**Figure 7.** Perturbation of proteins that regulate endoplasmic reticulum (ER) reorganization in aggregate clearance. (**A**) Images of cells expressing ER-FlucDM-eGFP after depletion of Atlastin 2 and Atlastin 3 and fixed at 65 min after released from the G2/M boundary. Maximum projected images were shown. (**B**) Quantification of cells expressing ER-FlucDM-eGFP after depletion of Atlastin 2 and Atlastin 3 and fixed at 65 min after released from the G2/M boundary. Control, 396 cells; ATL2/ATL3 KD, 362 cells. (**C**) Images of cells expressing ER-FlucDM-eGFP treated with DMSO or 100 nM MLN-8237

*Figure 7 continued on next page*

*Figure 7 continued*

and fixed at 65 min after released from the G2/M boundary. Maximum projected images were shown. (**D**) Quantification of cells expressing ER-FlucDM-eGFP treated with DMSO or 100 nM MLN-8237 and fixed at 65 min after released from the G2/M boundary. Only round mitotic cells were counted. DMSO, 102 cells; MLN-8237, 288 cells. (**E**) Images of cells expressing ER-FlucDM-eGFP treated with DMSO or 331 nM nocodazole and fixed at 65 min after released from the G2/M boundary. Maximum projected images were shown. (**F**) Quantification of cells expressing ER-FlucDM-eGFP treated with DMSO or 331 nM nocodazole and fixed at 65 min after released from the G2/M boundary. Only round mitotic cells were counted. DMSO, 122 cells; nocodazole, 320 cells. Scale bar, 10 μm. Data are from three independent experiments and analyzed by unpaired two-side Mann-Whitney U-test.

regulating ER-FlucDM-eGFP aggregate clearance appears indirect as ER-FlucDM-eGFP proteins are stable throughout cell division. It is possible that players involved in ER-FlucDM-eGFP aggregate clearance are regulated by proteasomal degradation, which are affected when proteasomes are inhibited. Interestingly, Melo et al. showed that aggregate clearance is promoted by treating ER-cells with ER stressors as the treatment triggers UPR, which in turn leads to higher activity of BiP to cope with the ER stress. We observed differential responses of aggregate clearance to ER stressors depending on the duration of the treatments. It is possible that acute treatment of the ER stressor on dividing cells does not provide sufficient time for cells to accumulate abundant amount of BiP given that mitotic cells are not very transcriptionally active (*Palozola et al., 2017*). Also, degradation of BiP could be accelerated upon the treatment of ER stressors, which in turn undermines the abundance of BiP in disaggregation (*Shim et al., 2018*). Thus, short-term treatment of the ER stressor in cells exacerbates the ER stress and affects aggregate clearance during cell division, whereas longer treatment of the ER stressor promotes aggregate clearance.

While the molecular mechanism by which the ER-FlucDM-eGFP is cleared during mitosis and cytokinesis remains to be elucidated, we favor a model in which proteins involved in the ER reorganization are regulated through protein modifications or proteasomal degradation to promote aggregate clearance in cell division. Also, BiP might function as a disaggregase to disassemble the ER-FlucDM-eGFP when cells exit mitosis. Furthermore, the pathological aggregates that are formed and cleared in a similar manner as the ER-FlucDM-eGFP are yet to be identified. We believe that identification of the pathological targets and mechanistic understanding of aggregate clearance during cell division could offer new insights to understand disaggregation in proteostasis control.

## Materials and methods

### Cell lines and culture

MCF10A cells were cultured in DMEM F-12 (Sigma) supplemented with 1% Glutamax (Gibco), 5% horse serum (Biological Industries), 20 ng/ml EGF (Gibco), 0.5 mg/ml Hydrocortisone (MedChemExpress), 100 ng/ml Cholera toxin (Sigma), 10 μg/ml Insulin (Biological Industries), and 1% Pen/Strep (Biological Industries). U2OS, A549, and HEK293Ta cells were maintained in DMEM (Sigma) with Glutamax (Gibco), 6% fetal bovine serum (Biological Industries), and 1% Pen/Strep (Biological Industries). MDA-MB-231 cells were cultured in DMEM F-12 (Sigma) with 1% Glutamax (Gibco) and 1% Pen/Strep (Biological Industries). All cell cultures were maintained at 37 °C with 5% $CO_2$ in a humidified incubator. All cells have been authenticated by Sangon (China) and frequently checked for mycoplasma contamination in the lab. Cells expressing ER-FlucDM-eGFP, ER-eGFP, ER-HaloDM-eGFP, ER-FlucWT-eGFP, and ER$^{FSHR}$-FlucDM-eGFP, ER-FlucDM-mCherry, NLS-FlucDM-eGFP, CFTR wild type, CFTR-ΔF508, AAT wild type, AAT S, and Z variants were generated through lentivirus transduction and selected with 5 μg/mL puromycin (Sangon Biotech). MCF10A cells expressing mStayGold-Sec61β were prepared by lentivirus transduction, selected with 50 μg/mL hygromycin B and only used in 3 d after transduction (*Ando et al., 2024*). MCF10A cells expressing CRISPRi plasmids to knock down ATL2 and ATL3 were selected by 6 μg/mL Blasticidin (Solarbio, 703X023) and only used in 5 d after transduction. MCF10A cells co-expressing ER-FlucDM-eGFP and ER-FlucDM-mCherry for FRAP or co-expressing ER-FlucDM-mCherry and mGreenLantern-hGeminin (degron) or co-expressing ER-FlucDM-mCherry and NLS-FlucDM-eGFP were generated by lentivirus transduction and selected by 5 ug/mL puromycin and followed by fluorescence-activated cell sorting (FACS) to enrich double-positive cells.

## Lentivirus packaging

Lentiviral particles were generated using the 3rd generation lentiviral packaging system. Packaging plasmids pRSV-Rev, pMDLg/pRRE, pMD2.G (kindly provided by Didier Trono), and the transfer plasmid were chemically transfected using GeneTwin transfection reagent (Biomed, TG101) into the HEK293Ta packaging cell line (Genecopoeia, LT008). The cell culture medium was harvested after 2 d and filtered through a 0.45 µm filter. The filtrate was concentrated using a 5 X precipitation solution (250 g/L PEG 8000 and 43.83 g/L NaCl in ddH$_2$O) overnight, and precipitated by centrifuging at 4000×g for 25 min at 4 °C. Lentiviral particles were resuspended in phosphate-buffered saline (PBS) and added to the culture medium, which were supplemented with 1 µg/mL polybrene (HanBio).

## Lentivirus expression constructs

Lentiviral transfer plasmids used in this study were listed below: pTGL0563 containing FlucDM-eGFP in which FlucDM was mutated from FlucWT (a gift from Dr. Mikael Bjorklund); pTGL0645 containing ER-FlucDM-eGFP-KDEL; pTGL0662 containing NLS-FlucDM-eGFP; pTGL0673 containing ER-FlucDM-mCherry-KDEL; pTGL0698 containing ER-FlucWT-eGFP-KDEL; pTGL0703 containing ER-HaloDM-eGFP-KDEL in which HaloDM was mutated from HaloWT; pTGL0707 containing ER-HaloWT-eGFP-KDEL; pTGL0733 containing ER-eGFP-KDEL; pTGL0743 containing ER$^{FSHR}$-FlucDM-eGFP; pTGL0737 containing CFTR-wt-eGFP; pTGL0738 containing CFTR-ΔF508-eGFP; pTGL0739 containing AAT-wt-eGFP; pTGL0740 containing AAT (S variant)-eGFP; pTGL0741 containing AAT (Z variant)-eGFP; pTGL0828 containing mGreenLantern-hGeminin (degron); pTGL0891 was the empty plasmid used in constructing pTGL0645 and used as a control for proteomics; pTGL0914 containing mStayGold-Sec61β; pTGL0386 was the empty plasmid in constructing CRISPRi plasmids; pCRISPRi0249 to knockdown ATL2 containing gRNA 5' GAGGGCAGCAACCGCACCAG 3'; pCRISPRi0252 to knock-down ATL3 containing gRNA 5' GAGCAGGGGTGCAGAGGAGA 3'. All plasmids are available upon request.

## Quantitative real-time PCR (qPCR)

Total RNA was extracted from cells using the FastPure Cell/Tissue Total RNA Isolation Kit V2 (Vazyme, RC112-01). Subsequently, the RNA was reversely transcribed into complementary DNA (cDNA) using the HiScript II Q RT SuperMix for qPCR (Vazyme, R223-01). The levels of the cDNAs were quantified using real-time PCR with the ChamQ Universal SYBR qPCR Master Mix (Vazyme, Q711-02). The PCR reaction mix was prepared on a hard-shell PCR plate (Bio-Rad, HSP9655), sealed with Microseal 'B' seals (Bio-Rad, MSB1001), and conducted in the CFX96 Touch Real-Time PCR Detection System (Bio-RAD, C1000). Primers used in the qPCR: GAPDH, 5' CAGGAGGCATTGCTGATGAT 3' and 5' GAAGGCTGGGGCTCATTT 3'; HSPA5, 5' CACAGTGGTGCCTACCAAGA 3' and 5' TGTCTTTTGTCA GGGGTCTTT 3'; Calreticulin, 5' ATAAAGGTTTGCAGACAAGC 3' and 5' CCACAGTCGATGTTCT GCTC 3'; HSP90B1, 5' TCCAGCAGAAAAGAGGCTGA 3' and 5' CAAATTCGGGAAGGGCCTGA 3'; CANX, 5' GCACCTATTCTGGAGGCGAG 3' and 5' ACAGCAACCACTTCCCTTCC 3'; P4HB, 5' TTCA GGAATGGAGACACGGC 3' and 5' TCCACGTCCTTGAAGAAGCC 3'; DNAJB9, 5' GTGGAGGA GCAGCAGTAGTC 3' and 5' CGCTCTGATGCCGATTTTGG 3'; ATL2, 5' CTGGTTCCATTGCTGCTTGC 3' and 5' CTTCAGCTGTTGCCTGAAGC 3'; ATL3, 5' TCACCCCAAGTCCATGCTTC 3' and 5' CTCC CCACAAACCTCTTCC 3'. The C$_T$ value for GAPDH was used for normalization to obtain the relative expression level.

## Protein immunoblotting

Adherent cells were lysed with 1 x LDS sample buffer (Beyotime) and boiled at 95 °C for 10 min. The 1 x LDS sample buffer was prepared by diluting 4 X LDS sample buffer (Beyotime) in 100 mM Tris-HCl (pH 7.4) with 2.5% v/v β-mercaptoethanol. Protein samples were separated by pre-cast SDS-PAGE gels (GenScript) and blotted to a PVDF membrane (EMD Millipore). Blots were blocked with Quick Block buffer (Beyotime) for at least 20 min at room temperature and were incubated at 4 °C overnight with the primary antibody. After rinsing blots with TBS containing 0.05% Tween-20 for four times (5 min each time), blots were incubated at room temperature with the secondary antibody for 1 hr, followed by washing with TBS containing 0.05% Tween-20. Odyssey CLX was employed to image the blots and LI-COR Image Studio software was used to quantify the band intensity. To incubate different primary antibodies in the same blot, the blot was vigorously rinsed with the stripping buffer (CWBIO,

01427/12724) and washed 3 times with TBS containing 0.05% Tween-20 and processed as described before.

To examine the solubility of ER-FlucDM-eGFP in *Figure 1—figure supplement 1E*, the lysis buffer containing 100 mM Tris-HCl (pH 7.4), 1% protease inhibitor cocktail (Lablead, C0101) and different concentrations of NP-40 (Sigma) was used. To prepare supernatant and pellet fractions of ER-FlucDM-eGFP, cells were lysed with the lysis buffer containing 0.2%, 0.5%, or 1.0% v/v NP-40. The lysate was then centrifuged at 14,000×g for 10 min at 4 °C and the supernatant was collected. The cell pellet was further lysed with 1 x LDS buffer on ice for 40 min and vortexed vigorously for 10 s every 10 min.

Primary antibodies used were: 1:3000 anti-GFP (Huabio, ET1601-13); 1:5000 anti-β-actin (Genscript, A00702); 1:1000 anti-Cyclin B1 (Proteintech, 55004–1-AP); 1:1000 anti-p53 (Huabio, ET1601-13). Secondary antibodies were goat anti-Rabbit IgG (H+L) Highly Cross-Adsorbed Secondary Antibody conjugated to Alexa Fluor Plus 800 (Invitrogen, A32735) and goat anti-Mouse IgG (H+L) Highly Cross-Adsorbed Secondary Antibody conjugated to Alexa Fluor Plus 680 (Invitrogen, A32729).

## Proteomics

MCF10A cells transduced with pTGL0891 (empty vector) or pTGL0645 (ER-FlucDM-eGFP) for 6 d were trypsinized, centrifuged, and washed twice with ice-cold PBS. The pellets were quickly frozen by liquid nitrogen and then sent to Tsingke (Wuhan, China) for further processing for proteomics. In brief, cell pellets were lysed with buffer containing 8 M urea, 1 mM PMSF, and 2 mM EDTA. Samples were subjected to ultrasound sonication on ice for 5 min and followed by centrifugation at 15,000 × g for 10 min. The supernatants were then collected, and their concentrations were measured using the BCA method (Beyotime). Solutions containing 100 μg was topped up to 200 μl with 8 M urea and DTT (final concentration 5 mM) was added before incubation at 37 °C for 45 min. Protein samples were alkylated using iodoacetamide (final concentration 11 mM) in a dark room at room temperature for 15 min. The solutions were then subjected to trypsin digestion and further analyzed using LC-MS/MS analysis. The MS/MS data were processed using DIA-NN (v1.8.1). The protein quantification of DIA-NN software is done by the MaxLFQ algorithm.

## Luciferase activity test

Each well of 96-well plates (Biosharp, BS-MP-96W for luminescence, BS-MP-96B for fluorescence) was seeded with 10,000 cells in 100 μL of medium. After incubation for 18 hr, 5 μM MG132 was added to the MG132 +FlucDM eGFP group, while other untreated wells remained unchanged. Following an additional 6 hr, cells were washed once with PBS.

For luminescence detection, cell lysis was achieved by adding 50 μL of Steady-Glo Luciferase Assay System buffer (Promega, E2510) to the wells, followed by a 15 min incubation at room temperature. Luminescence in each well was then recorded three times over 5 min using the Spark Multimode Microplate Reader (Tecan, 1000ms exposure). Luminescence intensity in each well was calculated by normalizing these three measurements and subtracting the background signal from empty wells.

To determine abundance of luciferase in each treatment, $1×10^5$ MCF10A cells were seeded in 12-well plates for 24 hr. The cells were lysed with 1 x LDS sample buffer (Beyotime) and protein immunoblotting using anti-β-actin and anti-GFP, respectively, was performed. β-actin was first used to normalize the GFP intensity, and the value was used to normalize the luminescence to calculate the luciferase activity.

## Mitotic cell preparation

To increase the quantity of dividing cells, cells were first seeded on imaging chambers (ibidi, 80826) and were synchronized to the G2/M boundary by treating cells with 7.5 μm RO-3306 (Selleckchem, S7747) for 16–20 hr. Cells were released to enter into mitosis synchronously by washing once with pre-warmed fresh medium.

## Drug treatments

The following drugs were used in this study: Thapsigargin (Abcam, ab120286), Tunicamycin (GlpBio, GC16738), MG132 (CSN Pharm, CSN11436), YUM70 (Aladdin, Y413413), VER-155008 (CSN Pharm, CSN13116), Apcin (Sigma, SML1503), proTAME (Cayman Chemical, 25835), Cycloheximide (AbMole, M4879), Nocodazole (Beyotime, S1765), MLN-8237 (TargetMol, T2241).

## Immunofluorescence staining

For HSPA5/BiP staining, MCF10A cells expressing ER-FlucDM-eGFP were seeded in 8-well ibidi plates, fixed with 100% methanol for 10 min and permeabilized with 0.5% Triton X-100 for 15 min. The fixed cells were blocked with 5% FBS in PBS for 30 min. Subsequently, cells were incubated with the mouse anti-BiP antibody (HuaBio, HA601076) at 1:200 dilution in 1% FBS overnight at 4 °C. The secondary antibody used was Donkey Anti-Mouse IgG H&L (Alexa Fluor 568, ab175472, Abcam) at 1:2000 dilution for 1 hr. For tubulin staining, cells were fixed using 4% PFA for 20 min and permeabilized with 0.5% Triton X-100 for 15 min. After blocking the fixed cells with 5% FBS in PBS for 30 min, cells were stained with the mouse anti-α-tubulin monoclonal antibody (Proteintech, 66031–1-IG) at 1:200 dilution. The secondary antibody used was Donkey Anti-Mouse IgG H&L (Alexa Fluor 568, ab175472, Abcam) at 1:2000 dilution for 1 hr. To stain DNA, DAPI Staining Solution (Sangon Biotech, E607303, 5 mg/L) was diluted in PBS to 50 ng/mL and applied to cells for 10 min. After the DAPI solution was removed, fixed cells were washed with PBS twice before imaging.

## Transmission electron microscopy (TEM)

To prepare samples for TEM, MCF10A cells expressing ER-FlucDM-eGFP were synchronized to the G2/M boundary by treating cells with RO-3306 for 16–20 hr and were released into mitosis by removing RO-3306. After 45 min released from the G2/M boundary, cells were trypsinized for 10 min before centrifugation. The cell pellet was fixed by 2.5% paraformaldehyde (in PBS) for 30 min at room temperature before overnight at 4 °C. Fixed pellet was further fixed by 1% osmium tetroxide and then 2% uranium acetate. Ethanol and acetone were used to dehydrate samples before embedding them into epoxy resin. Sectioned samples were observed by Tecnai G2 spirit 120 kV transmission electron microscopy (Thermo FEI).

## Live-cell staining

The live-cell staining dyes used are as follows: 100 nM SiR-Tubulin (Cytoskeleton, CY-SC002), 1 μM ER-tracker Red (Beyotime, C1041S) or 10 μM Thioflavin T (ThT) in the medium to label cells for about 30 min in the incubator before imaging. For ThT staining, about 10% of aggregates could be stained by ThT in ER-FlucDM-mCherry cells.

## Spinning-disk confocal microscopy imaging

The hardware configuration of the spinning-disk confocal microscopy system was as described previously (*Wang et al., 2023*). Imaging utilized a z-step size of 0.5 μm, with an x-y plane resolution of 183.3 nm/pixel for the 60x lens and 110 nm/pixel for the 100x lens. Fluorophores were excited using laser lines at wavelengths of 405, 488, 561, or 640 nm.

For FRAP analysis, images were acquired with 488 nm and 561 nm laser lines through the 100x lens. The procedure involved acquiring three pre-bleach images every 30 s, followed by bleaching the mCherry signal using the 561 nm laser. Subsequently, one image was acquired immediately post-bleaching, followed by ten images acquired every 30 s. In total, fourteen time-point images were acquired, and the total duration of each FRAP experiment was 6.5 min.

## Image analysis and processing

Fiji was used to process images. Z-stack images for quantification were first projected using max-intensity projection (Fiji/image/stacks/Z project). To quantify aggregates, projected images were classified by Trainable Weka Segmentation (Fiji/Plugins/Segmentation/Trainable Weka Segmentation). The segmented images were processed by using the ImageJ Macro commands: run('8-bit'); setOption('BlackBackground,' true); run('Convert to Mask'); run('Watershed') before using Analyze Particles (Fiji/Analyze/Analyze Particles). For time-lapse images except for images from FRAP, the segmented images were processed by using the ImageJ Macro commands: run('8-bit'); setThreshold(0, 120); run('Make Binary,' 'method = Default background = Light black'); run('Invert LUT'); run('Watershed,' 'stack') before using Analyze Particles (Fiji/Analyze/Analyze Particles).

To quantify total GFP intensities in live cells, Z-stack images (24 μm) were acquired and projected using sum slices projection. Cell outlines were segmented using polygon selection in Fiji. To quantitate the background intensity, the mean intensity of a small dark region of the field of view was

measured and multiplied by the area of the segmented cell. The background was subtracted from the GFP intensity of segmented cells.

For FRAP analysis of ER-eGFP and ER-HaloDM-eGFP, regions of interest (ROI) in projected images were manually selected and the average fluorescence intensity of ROIs was measured in Fiji. For FRAP analysis of aggregates in cells co-expressing ER-FlucDM-eGFP and ER-FlucDM-mCherry (both interphase and mitosis), the aggregates were first segmented by Weka Segmentation. The unsegmented images from the mCherry channel were processed by Analyze Particles (Fiji/Analyze/Analyze Particles) based on the corresponding segmented images from the GFP channel. For fluorescence intensity quantification, background in all FRAP experiments were subtracted by one hundred.

## Quantification and statistical analysis

Datasets were analyzed by Student's t-test or Mann-Whitney test. Prism 8 (GraphPad) was used for the statistical analysis and data plotting. Statistical details are indicated in the figure legends.

## Acknowledgements

This study was funded by the National Natural Science Foundation of China (NSFC) RFIS-II grant (32350610247) and NFSC grant (32270770) given to TGC We thank Xiaoxia Wan and Chenyu Yang in the Center of Cryo-Electron Microscopy (CCEM), Zhejiang University for their technical assistance on Transmission Electron Microscopy. We thank Xuqi Chen, Xukai Gao, and Zihan Chu for participating in the early stage of the project. We thank Dr. Mike Shipston for participating in the discussion. We thank all lab members for the discussion.

## Additional information

### Funding

| Funder | Grant reference number | Author |
| --- | --- | --- |
| National Natural Science Foundation of China | 32350610247) | Ting Gang Chew |
| National Natural Science Foundation of China | 32270770) | Ting Gang Chew |

The funders had no role in study design, data collection and interpretation, or the decision to submit the work for publication.

### Author contributions

Shoukang Du, Conceptualization, Formal analysis, Investigation, Methodology, Writing – original draft, Writing – review and editing; Yuhan Wang, Bowen Chen, Shuangshuang Xie, Kuan Yoow Chan, Resources; David C Hay, Supervision, Writing – review and editing; Ting Gang Chew, Conceptualization, Resources, Supervision, Writing – original draft, Project administration, Writing – review and editing

### Author ORCIDs

Shoukang Du ![ORCID] https://orcid.org/0000-0001-5618-947X
Ting Gang Chew ![ORCID] https://orcid.org/0000-0001-5902-3897

Reviewer #1 (Public review): https://doi.org/10.7554/eLife.96675.4.sa1
Reviewer #2 (Public review): https://doi.org/10.7554/eLife.96675.4.sa2
Reviewer #3 (Public review): https://doi.org/10.7554/eLife.96675.4.sa3
Author response https://doi.org/10.7554/eLife.96675.4.sa4

## Additional files

**Supplementary files**
MDAR checklist

**Data availability**
All data generated or analysed during this study are included in the manuscript and supporting files; source data files containing the blots have been provided for Figure 1-figure supplement 1 and Figure 6-figure supplement 2.

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
