## [Editor Report · eLife Assessment]

How misfolded proteins are segregated and cleared is a significant question in cell biology, since clearance of these aggregates can protect against pathologies that may otherwise arise. The authors discover a cell cycle stage-dependent clearing mechanism that involves the ER chaperone BiP, the proteosome, and CDK inactivation, but is curiously independent of the anaphase promoting complex (APC). These are **valuable** and interesting new observations, and the evidence supporting these claims is **solid**.

---

## [Referee Report · Reviewer #1 (Public review)]

Du et al. address the cell cycle-dependent clearance of misfolded protein aggregates mediated by the endoplasmic reticulum (ER) associated Hsp70 chaperone family and ER reorganisation. The observations are interesting and impactful to the field.

Strength:

The manuscript addresses the connection between the clearance of misfolded protein aggregates and the cell cycle using a proteostasis reporter targeted to ER in multiple cell lines. Through imaging and some biochemical assays, they establish the role of BiP, an Hsp70 family chaperone, and Cdk1 inactivation in aggregate clearance upon mitotic exit. Furthermore, the authors present an initial analysis of the role of ER reorganisation in this clearance. These are important correlations and could have implications for ageing-associated pathologies. Overall, the results are convincing and impactful to the field.

Weakness:

The manuscript still lacks a mechanistic understanding of aggregate clearance. Even though the authors have provided the role of different cellular components, such as BiP, Cdk1 and ATL2/3 through specific inhibitors, at least an outline establishing the sequence of events leading to clearance is missing. Moreover, the authors show that the levels of ER-FlucDM-eGFP do not change significantly throughout the cell cycle, indicating that protein degradation is not in play. Therefore, addressing/elaborating on the mechanism of disassembly can add value to the work. Also, the physiological relevance of aggregate clearance upon mitotic exit has not been tested, nor have the cellular targets of this mode of clearance been identified or discussed.

---

## [Referee Report · Reviewer #2 (Public review)]

This paper describes an interesting observation that ER-targeted misfolded proteins are trapped within vesicles inside nucleus to facilitate quality control during cell division. This work supports the concept that transient sequestration of misfolded proteins is a fundamental mechanism of protein quality control. The authors satisfactorily addressed several points asked in the review of first submission. The manuscript is improved but still unable to fully address the mechanisms.

Strengths:

The observations in this manuscript are very interesting and open up many questions on proteostasis biology.

Weaknesses:

Despite inclusions of several protein-level experiments, the manuscript remained a microscopy-driven work and missed the opportunity to work out the mechanisms behind the observations.

---

## [Referee Report · Reviewer #3 (Public review)]

This paper describes a new mechanism for the clearance of protein aggregates associated to endoplasmic reticulum re-organization that occurs during mitosis.

Experimental data showing clearance of protein aggregates during mitosis is solid, statistically significant, and very interesting. The authors made several new experiments included in the revised version to address the concerns raised by reviewers. A new proteomic analysis, co-localization of the aggregates with the ER membrane Sec61beta protein, expression of the aggregate-prone protein in the nucleus does not result in accumulation of aggregates, detection of protein aggregates in the insoluble faction after cell disruption and mostly importantly knockdown of ATL proteins involved in the organization of ER shape and structure impaired the clearance mechanism. This last observation addresses one of the weakest points of the original version which was the lack of experimental correlation between ER structure capability to re-shape and the clearance mechanism.

In conclusion, this new mechanism of protein aggregate clearance from the ER was not completely understood in this work but the manuscript presented, particularly in the revised version, an ensemble of solid observations and mechanistic information to scaffold future studies that clarify more details of this mechanism. As stated by the authors: "How protein aggregates are targeted and assembled into the intranuclear membranous structure waits for future investigation". This new mechanism of aggregate clearance from the ER is not expected to be fully understood in a single work but this paper may constitute one step to better comprehend the cell capability to resolve protein aggregates in different cell compartments.

[Editors' note: The authors have appropriately addressed the previous reviewers' concerns.]

---

## [Author Response]

The following is the authors’ response to the previous reviews

**Public Reviews:**

**Reviewer #1 (Public review):**
Du et al. address the cell cycle-dependent clearance of misfolded protein aggregates mediated by the endoplasmic reticulum (ER) associated Hsp70 chaperone family and ER reorganisation. The observations are interesting and impactful to the field.Strength:The manuscript addresses the connection between the clearance of misfolded protein aggregates and the cell cycle using a proteostasis reporter targeted to ER in multiple cell lines. Through imaging and some biochemical assays, they establish the role of BiP, anHsp70 family chaperone, and Cdk1 inactivation in aggregate clearance upon mitotic exit.Furthermore, the authors present an initial analysis of the role of ER reorganisation in this clearance. These are important correlations and could have implications for ageingassociated pathologies. Overall, the results are convincing and impactful to the field.Weakness:The manuscript still lacks a mechanistic understanding of aggregate clearance. Even though the authors have provided the role of different cellular components, such as BiP, Cdk1 and ATL2/3 through specific inhibitors, at least an outline establishing the sequence of events leading to clearance is missing. Moreover, the authors show that the levels of ERFlucDM-eGFP do not change significantly throughout the cell cycle, indicating that protein degradation is not in play. Therefore, addressing/elaborating on the mechanism of disassembly can add value to the work. Also, the physiological relevance of aggregate clearance upon mitotic exit has not been tested, nor have the cellular targets of this mode of clearance been identified or discussed.

Thank you for your suggestions.

We have added descriptions about the sequence of events leading to clearance in the abstract (line 33) and discussion (line 316).

We have commented on the future work that could address the molecular mechanisms behind the aggregate clearance in the discussion (line 388).

It has been difficult to address the physiological relevance of aggregate clearance during cell division, as the inhibition of BiP or depletion of ATL2/3 that prevent aggregate clearance cause cellular consequences not specific to aggregate clearance. Future work that lead to understanding of aggregate clearance at the molecular level may allow us to address this more specifically. Furthermore, we have commented about the potential defects that could arise in cells expressing ER-FlucDM-eGFP that have a perturbed cellular health based on the proteomic analysis (line 359).

To identify pathological targets that undergo clearance as the ER-FlucDM-eGFP, we tested three pathological mutants (CFTR-∆F508, AAT S and Z variants) that are known to mis-fold and accumulate in the ER. Unfortunately, expression of these mutants did not result in the confinement of aggregates in the nucleus. The data related to this have been added as Figure S1E and S1F (line 102) in this revised manuscript. We have also commented in the discussion that pathological targets are yet to be identified and could be a part of future work (line 392).

**Reviewer #2 (Public review):**
This paper describes an interesting observation that ER-targeted misfolded proteins are trapped within vesicles inside nucleus to facilitate quality control during cell division. This work supports the concept that transient sequestration of misfolded proteins is a fundamental mechanism of protein quality control. The authors satisfactorily addressed several points asked in the review of first submission. The manuscript is improved but still unable to fully address the mechanisms.Strengths:The observations in this manuscript are very interesting and open up many questions on proteostasis biology.Weaknesses:Despite inclusions of several protein-level experiments, the manuscript remained a microscopy-driven work and missed the opportunity to work out the mechanisms behind the observations.

Thank you for your suggestions. We believe that our study has provided a genetic basis for the involvement of ER reorganization and BiP during cell division in aggregate clearance, which is a new observation. We have also commented in this revised manuscript about the future work that could address the molecular mechanisms behind the aggregate clearance in the discussion (line 388).

**Reviewer #3 (Public review):**
This paper describes a new mechanism for the clearance of protein aggregates associated to endoplasmic reticulum re-organization that occurs during mitosis.Experimental data showing clearance of protein aggregates during mitosis is solid, statistically significant, and very interesting. The authors made several new experiments included in the revised version to address the concerns raised by reviewers. A new proteomic analysis, co-localization of the aggregates with the ER membrane Sec61beta protein, expression of the aggregate-prone protein in the nucleus does not result in accumulation of aggregates, detection of protein aggregates in the insoluble faction after cell disruption and mostly importantly knockdown of ATL proteins involved in the organization of ER shape and structure impaired the clearance mechanism. This last observation addresses one of the weakest points of the original version which was the lack of experimental correlation between ER structure capability to re-shape and the clearance mechanism.In conclusion, this new mechanism of protein aggregate clearance from the ER was not completely understood in this work but the manuscript presented, particularly in the revised version, an ensemble of solid observations and mechanistic information to scaffold future studies that clarify more details of this mechanism. As stated by the authors: "How protein aggregates are targeted and assembled into the intranuclear membranous structure waits for future investigation". This new mechanism of aggregate clearance from the ER is not expected to be fully understood in a single work but this paper may constitute one step to better comprehend the cell capability to resolve protein aggregates in different cell compartments.

We thank the reviewer for the comments.

**Recommendations for the authors:**

**Reviewer #1 (Recommendations for the authors):**
The manuscript presents a very interesting set of observations that could have significant implications on age-related protein misfolding and aggregate clearance. There are a few places in the manuscript that still need more clarity. Some are listed below, which I think can improve the manuscript.- The new data associated with proteomic analysis is appreciated, but the information gained has not been explored or elaborated sufficiently in the manuscript. Based on the differential expression of cell cycle proteins, how the authors interpret cellular health is unclear. Also, the physiological role of this mode of aggregate clearance remains unclear.

We have added our interpretation of perturbed cellular health in cells expressing ERFlucDM-eGFP in the discussion (line 359).

It has been difficult to address the physiological relevance of aggregate clearance during cell division, as the inhibition of BiP or depletion of ATL2/3 that prevent aggregate clearance cause cellular consequences not specific to aggregate clearance. Future work that lead to understanding of aggregate clearance at the molecular level may allow us to address this more specifically.

- In Figure 3A, have the authors measured the total GFP intensity from interphase through early G1? Even though the number and area of the aggregates decrease significantly, the cytoplasmic GFP signal does not seem to increase. Considering new CHX chase experiments and total Fluorescence intensity calculations (Figure S7D), which indicate no difference, one would expect an increase in cytoplasmic signal upon the disassembly of aggregates. Therefore, the data from Figures 3A and 7D seem contradictory. Can the authors please explain?

We apologized for the confusion. The images in Figure 3A were derived from fixed cells. So, different cells were shown in every cell cycle phases and were not suitable for quantification. Fluorescence intensity changes could be better appreciated in Figure 3C or 4D as these were time-lapse microscopy images of live cells progressing through mitosis and cytokinesis. Data used in the quantification of fluorescence intensity in Figure S7D were derived from live cells taken from specific time points to avoid unwanted fluorescence bleaching during time-lapse microscopy.

- Do the authors expect a similar clearance of pathological aggregates such as mutant FUS or TDP43 condensates? Showing aggregate disassembly of disease-relevant aggregates would be an excellent addition to the manuscript, but it might be beyond the scope of the current version. However, the authors can comment/speculate how their study might extend to pathological condensates.

We tested three pathological mutants (CFTR-∆F508, AAT S and Z variants) that are known to mis-fold and accumulate in the ER. Unfortunately, expression of these mutants did not result in the confinement of aggregates in the nucleus. The data related to this have been added as Figure S1E and S1F (line 102) in this revised manuscript. We have commented that pathological targets are yet to be identified and could be a part of future work (line 392).

- The presence of ER membrane around these aggregates is an interesting observation. This membrane is retained even after nuclear membrane breakdown. What could be the relevance of membrane-bound aggregates, especially since the membrane can limit the access of chaperones involved in disassembly? This observation becomes more important since the depletion of ER membrane fusion proteins also leads to the accumulation of aggregates. Are the membranes a beacon for disassembly? The authors may comment/ speculate. This could also be an important aspect of the mechanism of clearance.

We think that the ER membranes around the aggregates are disassembled when the ER networks reorganize during mitotic exit and this may allow accessibility of BiP to disaggregate the aggregates. We have added this in the discussion (line 316).